# ampycloud: an open-source algorithm to determine cloud base heights and sky coverage fractions from ceilometer data

Frédéric P.A. Vogt[1], Loris Foresti[2], Daniel Regenass[1], Sophie Réthoré[3], Néstor Tarin Burriel[3], Mervyn Bibby[3], Przemysław Juda[2], Simone Balmelli[2], Tobias Hanselmann[3], Pieter du Preez[3], and Dirk Furrer[3]

[1]Federal Office of Meteorology and Climatology - MeteoSwiss, Chemin de l'Aérologie 1, 1530 Payerne, Switzerland.
[2]Federal Office of Meteorology and Climatology - MeteoSwiss, Via ai Monti 146a, 6605 Locarno-Monti, Switzerland.
[3]Federal Office of Meteorology and Climatology - MeteoSwiss, Operation Center 1, 8058 Zurich-Airport, Switzerland.

**Correspondence:** Frédéric P.A. Vogt (frederic.vogt@meteoswiss.ch)

**Abstract.** Ceilometers are used routinely at aerodromes worldwide to derive the height and sky coverage fraction of cloud layers. This information, possibly combined with direct observations by human observers, contributes to the production of Meteorological Aerodrome Reports (METARs). Here, we present ampycloud, a new algorithm and associated Python package for automatic processing of ceilometer data, with the aim to determine the sky coverage fraction and base height of cloud layers above aerodromes. The ampycloud algorithm has been developed at the Swiss Federal Office of Meteorology and Climatology (MeteoSwiss) as part of the AMAROC (AutoMETAR/AutoReport rOund the Clock) program, to help in the fully automatic production of METARs at Swiss civil aerodromes. ampycloud is designed to work with no direct human supervision. The algorithm consists of three distinct, sequential steps that rely on agglomerative clustering methods and Gaussian mixture models to identify distinct cloud layers from individual cloud base hits reported by ceilometers. The robustness of the ampycloud algorithm stems from the first processing step, simple and reliable. It constrains the two subsequent processing steps that are more sensitive, but also better suited to handle complex cloud distributions. The software implementation of the ampycloud algorithm takes the form of an eponymous, pip-installable Python package developed on GitHub and made publicly accessible.

## 1 Introduction

Ceilometers are being used at numerous aerodromes worldwide to derive autonomously the base height and sky coverage fraction of cloud layers (Wauben et al., 2006; ICAO, 2011; de Haij et al., 2016). These parameters form an essential component of METerological Aerodrome Reports (METARs; WMO, 2022). These compact telegrams provide a detailed description of the current meteorological conditions at and around the aerodrome ground to pilots, air traffic controllers, and aerodrome safety services. Regulations from the International Civil Aviation Organization (ICAO) and the corresponding European Commission of Implementing Regulation (CIR, EU373) dictate that METARs should be issued at 30 min intervals, or less in case of a rapid change of operational significance in the meteorological conditions. Depending on local agreements, the latter situation leads to a SPECI (with the same template as METARs) or a local special report (SPECIAL; with the same template as local routine reports, known as MET REPORTs). METARs/SPECIs are representative of the whole aerodrome and are thus mainly used

by pilots for flight planning, while MET REPORTs/SPECIALs are representative of the threshold and touchdown zone of a specific runway, and are mostly used for managing landing and departure operations by air traffic controllers (ATCs).

The first METAR in Switzerland was issued in 1969 (Willemse and Furger, 2017). The Swiss Federal Office of Meteorology and Climatology (MeteoSwiss) is currently responsible for the production of METARs at the Swiss civil aerodromes. These include the international aerodromes of Geneva (ICAO code: LSGG) and Kloten (ICAO code: LSZH), and 8 regional aerodromes. Since 2007, it has done so using the SM⋀RT (System for Meteorological Automated ReporTing) software that

has been fully developed and maintained by MeteoSwiss. SM⋀RT is comprised of both a backend component and a frontend component. The backend is responsible for the data collection from meteorological sensors and includes algorithms for the generation of METAR/MET REPORT/SPECIAL Proposals every minute. The frontend includes the SM⋀RT editor which is used by the aeronautical meteorological observers (AMOs) to compile and send the reports (supported by the message Proposals), and a series of "viewers" showing for example the real-time data from meteorological sensors at different runways,

among others.

The 24/7 automation of METARs is a challenging objective that has been pursued by several meteorological services over the years (see e.g. Leroy, 2006; Wauben et al., 2006; Hartley and Quayle, 2014; JMA, 2022). With its AMAROC (AutoMETAR/AutoReport rOund the Clock) program, MeteoSwiss is no exception (MeteoSwiss, 2022). At first, the automatic

METAR, MET REPORT and SPECIAL Proposals generated by SM⋀RT at LSGG and LSZH have been systematically reviewed and adjusted by AMOs during the aerodrome's operational hours (05:30-23:30 local time). Since November 2016 at LSGG and April 2021 at LSZH, during non-operational hours only, SM⋀RT-generated METARs have been issued without any human interaction. At LSGG, they are now being distributed around the clock since 2024-05-01. These are referred to as AUTO METARs, AUTO MET REPORTs and AUTO SPECIALs.


Deriving cloud base heights and sky coverage fractions from ceilometer data requires dedicated software. A pioneering algorithm assembled by Esbjörn Olsson at the Swedish Meteorological and Hydrological Institute (SMHI) in 1995 was subsequently shared with ceilometer manufacturers Vaisala and Eliasson (at least), who further developed it (B. Nordberg, SMHI, private communication). The original SMHI algorithm was also the base for the software deployed at aerodromes in the

Netherlands by the Royal Netherlands Meteorological Institute (KNMI) (Wauben, 2002). In the Unites States, the "sky condition algorithm" developed for the Automated Surface Observing Systems (ASOS; Nadolski, 1998) is another example.

Despite their widespread use, very little detailed information exists on these different algorithms, beyond the fact that they rely on clustering methods. None of the associated software are open-source, and several are in fact considered trade secrets.

As a result, National Weather Services who want to have full control on their software are often forced to re-develop custom algorithms and/or associated software from scratch. The opacity surrounding the different algorithms responsible for generating AUTO METARs worldwide prevents any external (neutral) assessment, validation, and/or intercomparison of their capabilities.

Furthermore, the lack of open-source software and publicly-available documentation hinders the improvement of algorithms through collaborative work. This opacity may also impede the combined use of distinct ceilometer types at a given site.

In this article, we present ampycloud, a new open-source algorithm and associated Python package designed to derive the base height and sky coverage fraction of cloud layers from ceilometer data. ampycloud is part of the SM∧RT backend. It is one among several new and/or improved algorithms developed at MeteoSwiss as part of the AMAROC program, with the goal to generate fully-automated METAR reports at Swiss civil aerodromes. The algorithm itself is introduced in Sec. 2. The ampycloud Python package, its computational performance and accuracy, as well as its deployment and operational use by MeteoSwiss are all described in Sec. 3. The known limitations of ampycloud are discussed in Sec. 4, together with different enhancement possibilities. Our conclusions are presented in Sec. 5.

## 2 The ampycloud algorithm

### 2.1 Requirements

An algorithm that has to operate 24/7 without direct human supervision should meet specific requirements. In particular, it should be:

1. robust (any input, no matter its plausibility, is processable), and

2. reliable (any input results in a reasonable output).

For the specific case of generating information to be used in METARs, the algorithm should also be designed to be:

3. stable (a small change in the input results in a small change in the output),

4. conservative (if two possible outputs are equally probable, the worst one – from an aerodrome/flight safety perspective – is to be favored), and

5. fast (the processing time should be less than ∼1 min, to issue SPECIs or SPECIALs as soon as warranted).

Furthermore, any meteorological algorithm can also benefit from being:

6. physics-based (methods and parameters have a physical interpretation and justification).

This can ease, for example, the selection of parameter values and the identification of specific shortcomings of the algorithm, and thus elements with a potential for improvement. A supervised machine learning method trained to reproduce (subjective) cloud observations would be an example of a non-physics-based algorithm, which is generally more difficult to interpret and often expensive to maintain (typically denoted as a black box; see e.g. Sculley et al., 2015; Rudin, 2019). ampycloud was

developed with these different requirements in mind.

One should note that the ampycloud algorithm was not developed because the other algorithms serving similar goals used at aerodromes worldwide are unfit-for-purpose. Rather, it is the lack of information and transparency about their design that led MeteoSwiss to assemble ampycloud, also to ensure its long-term maintainability and extendability.

## 2.2 Algorithm concept

The ampycloud algorithm is composed of three sequential steps, which we refer to as *slicing*, *grouping*, and *layering*. The slicing step is designed for robust detection of the main cloud layers. It is meant to constrain the ampycloud outputs to a reasonable range, no matter the input data. The high degree of reliability of this step is achieved by reducing its capacity to handle complex cloud distributions. Specifically, the slicing step is subject (by design) to two pitfalls:

1. it can split cloud layers that span a broad range of heights and/or have a marked up-/downward trend, or

2. it can aggregate close-but-distinct cloud layers.

The grouping and layering steps of the ampycloud algorithm are designed specifically to refine the slicing step. The slicing, grouping, and layering steps of ampycloud thus form a coherent sequence, with each step operating on the outcome of the previous one.

ampycloud reports compliant METAR codes[1] of up to 3 cloud layers at most, with the following selection rules (ICAO, 2018):

– the lowest layer is always reported

– the second layer up is reported if it has a sky coverage of SCT or more

– the third layer up is reported if it has a sky coverage of BKN or more.

It must be stressed that when used on its own, ampycloud can comply with only a subset of the ICAO's rules for cloud reporting, in the sense that it is designed to characterize cloud layers with well-defined bases only. The ampycloud algorithm does not concern itself with the question of whether a vertical visibility (VV) code should be reported instead of a cloud base height in METARs. At MeteoSwiss, this aspect is being handled by different tools within the SM⋀RT infrastructure. Obscuration of the sky by fog or snow is also being handled by a separate algorithm within SM⋀RT. ampycloud does not consider the question of convective cloud detection (CB/TCU) either, which is best done using a combination of lightning and weather radar data. Regulations state that "*if there are no clouds of operational significance and no restriction on vertical visibility and the abbreviation CAVOK is not appropriate, the abbreviation NSC should be used*" (ICAO, 2018). Given its scope, ampycloud

---

[1]By definition, the METAR codes FEW, SCT, BKN, and OVC correspond to sky coverage fractions of 1-2 oktas, 3-4 oktas, 5-7 oktas, and exactly 8 oktas (respectively).

cannot decide whether a CAVOK code is appropriate, and will therefore always return NSC if there are no cloud layers found below the Minimum Sector Altitude (MSA). If no clouds are detected at all by the ceilometers, ampycloud will return NCD. Importantly, users should bear in mind that because ampycloud does not/cannot handle CB and TCU cases, any NCD/NSC code issued by ampycloud may need to be overwritten by the user in certain situations.

## 2.3   Input data

The World Meteorological Organization (WMO) defines a cloud base as "*a zone in which the obscuration corresponding to a change from clear air or haze to water droplets or ice crystals causes significant changes in the profiles of the backscatter and extinction coefficients*" (WMO, 2021). Ceilometers deployed at aerodromes worldwide are designed to detect such changes in backscatter coefficient profiles (ICAO, 2011), that are typically reported in the form of cloud base and/or vertical visibility (VV) hits (see e.g. Vaisala Oyj, 2015; Campbell Scientific, 2021; OTT HydroMet, 2022), but with significant differences be-

tween ceilometer types/models (Martucci et al., 2010; Görsdorf et al., 2016, 2018). ampycloud is designed to use cloud and VV hits acquired by one or more ceilometers over a time interval $\Delta t$ to characterize the cloud base heights and sky coverage fractions above a specific area. This is a well-established approach (Wauben et al., 2006; Nadolski, 1998; Wauben et al., 2006; ICAO, 2011; WMO, 2021). It relies on the wind *dragging* the cloud distribution over the ceilometer(s) line(s)-of-sight to enhance their otherwise-very-restricted spatial sampling capabilities, and thus obtain a more representative view of the sky. The

longer the time interval or the larger the wind speed, the better the spatial representativity of the dataset, but the worse is the view of the "current" state of the sky (especially in case of rapidly changing conditions).

ampycloud requires every cloud and VV hit $h$ to be comprised of four elements:

$$h \equiv (h_{\mathrm{hgt}}, h_{\mathrm{tme}}, h_{\mathrm{cid}}, h_{\mathrm{tpe}}), \tag{1}$$

with $h_{\mathrm{hgt}}$ the cloud base height above aerodrome level (aal; in ft), $h_{\mathrm{tme}}$ the observation time (in seconds, either as absolute Unix time or relative to a specific event, e.g. the most recent ceilometer data in the set), $h_{\mathrm{cid}}$ a ceilometer reference id, and $h_{\mathrm{typ}}$ the hit type, that will be explained below. The model and amount of ceilometers in operation at LSGG and LSZH is summarized in Table 1. The use of ampycloud is not restricted to any specific model of ceilometer in particular, provided that individual hits can be specified following Eq. 1. One should note that for the moment, ampycloud cannot use full backscatter

profiles to derive cloud base hits independently from the ceilometers' proprietary software. Implementing such a capability could enhance the robustness and traceability of the hit derivation, but it would not fundamentally change the behavior of the ampycloud algorithm in itself.

ampycloud can process measurements from any number of ceilometers, and makes no distinction between them. Data from

different ceilometers are analyzed together, independently of the spatial distribution of the instruments on the ground. The specification of $h_{\mathrm{cid}}$ for every hit ensures that a correct estimation of the sky coverage fraction can be made, under the assumption that cloud layers have a unique base per time step, per ceilometer line-of-sight. Without a value of $h_{\mathrm{cid}}$ to differentiate

**Table 1.** Summary of the ceilometers deployed at the LSGG and LSZH civil aerodromes, as of May 2024. The MSA values are expressed in ft above aerodrome level (ft aal; MeteoSwiss, 2023).

| Aerodrome | City | Elevation | MSA | Ceilometer | | | |
|---|---|---|---|---|---|---|---|
| | | | | count | model | manufacturer | measurement time interval |
| LSGG | Geneva | 1'411 ft amsl | 10'000 ft aal | 4 | CL31 | Vaisala | 15 s |
| LSZH | Kloten | 1'416 ft amsl | 8'000 ft aal | 7 | CL31 | Vaisala | 15 s |

sightlines, two simultaneous measurements from distinct ceilometers would not be distinguishable, and thus possibly become contradictory. One can of course apply ampycloud on a smaller number of ceilometers, e.g. on a specific subset associated to a given runway to generate local routine reports (AUTO MET REPORTs).

Aside from operational redundancy, one benefit of using multiple ceilometers to characterize cloud layers lies in the boost of the probability to detect clouds (resp. clear sky) for cloud layers with very low (resp. very high) sky coverage fractions. This fact, demonstrated experimentally by Wauben (2002), is illustrated in Fig. 1. The ability to better distinguish between OVC and BKN layers may not be a crucial benefit in itself, given that both categories imply the existence of an (operationally-relevant) "ceiling". However, using multiple ceilometers can also contribute to reducing cases of gross over-/underestimation of sky coverage fractions in slow-moving conditions (i.e. low-wind with quasi-stationary clouds), provided that the horizontal separation between ceilometers is larger than the applicable cloud characteristic scale length (Slobodda et al., 2015; Denby et al., 2022).

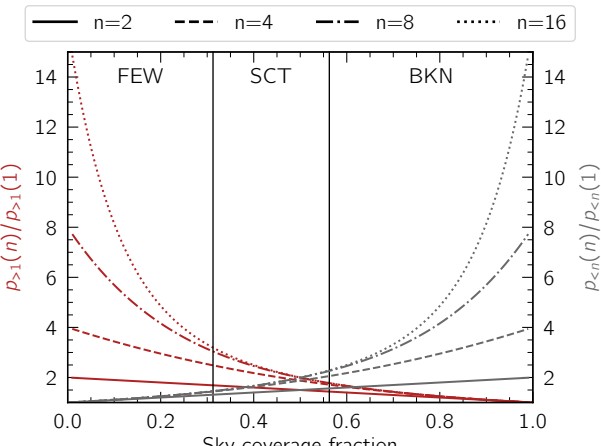

**Figure 1.** Theoretical improvement in the probability of getting at least one cloud hit with $n$ ceilometers $p_{>1}(n)$ compared to only one ceilometer $p_{>1}(1)$ (at any one specific moment in time), ignoring any spatial scale considerations in the cloud and ceilometer distributions (red curves). The mirrored probabilities of detecting at least one clear sight-line $p_{<n}$ are shown in gray.

It is important to consider the hit type $h_{\mathrm{tpe}}$ to derive a correct estimation of the sky coverage fraction of cloud layers. Typically, ceilometers can report multiple cloud base heights and/or a vertical visibilities with every observation (i.e. at every time step). For example, the ceilometers deployed at LSZH and LSGG report, for every measurement, either up to 3 distinct cloud base heights, or a vertical visibility and signal range (Vaisala Oyj, 2015). The latter occurs if the sky is obscured (for example due to fog or precipitation). The hit type $h_{\mathrm{tpe}}$ is used to keep track of these differences. It must be stressed, however, that

ampycloud does not treat vertical visibility (VV) hits differently than regular hits. Every hit $h$ inputted to ampycloud is treated as a regular cloud base hit. It is up to the user to decide if and when VV hits should be provided to ampycloud, and whether they need to be pre-processed in any way before doing so. Not passing VV hits to ampycloud can lead to an underestimation of the sky coverage fraction. However, VV hits can also be caused by precipitation that might lead to the reporting of spurious (lower) cloud layers. As of ampycloud v1.0.0, MeteoSwiss inputs any VV hits reported by CL31 ceilometers to the code,

without further selection or modification of their reported height.

     ampycloud works in a relative time space, and will automatically assign to each measurement a time difference with respect to the latest one. The algorithm poses no restriction on the time interval $\Delta t$ over which cloud and VV hits can be bundled. For historical reasons, cloud and VV hits spanning $\Delta t = 15\,\mathrm{min}$ for AUTO METARs (from all ceilometers) and $\Delta t = 6\,\mathrm{min}$

for AUTO MET REPORTS (from specific pairs of ceilometers associated to a given runway) serve as input for ampycloud at LSGG and LSZH. Users can however freely choose to collect cloud and VV hits over a longer time interval: ampycloud will process whatever collection of cloud and VV hits it receives as input.

### 2.4    The parameters of ampycloud

The parameters of ampycloud are summarized in Table 2. Those specifically related to the slicing, grouping, and layering

steps will be discussed in Sec. 2.6. The other ampycloud parameters are responsible for the generation of compliant METAR codes. It must be stressed that the default values provided in Table 2 for the different parameters are not necessarily applicable universally. Depending on their needs and/or instrumental setups, some users may wish to consider adjusting some of them, for example $\Theta_0$ and/or $\Theta_8$. $\Theta_0$ is the maximum (absolute) number of cloud hits for a given slice/group/layer to still be considered to be NCD by ampycloud; $\Theta_8$, on the other hand, is the maximum number of "holes" (i.e. non-detection of clouds) for a given

slice/group/layer to be considered to be OVC by ampycloud. We adopt $\Theta_0 = 3$ hits ($\Theta_8 = 1$ hole) at LSGG to avoid assigning cloud layers with low (high) sky coverage fractions to noise fluctuations and false cloud detections (e.g. returns by airplanes passing above a ceilometer[2]).

     ampycloud computes the base height of a given slice $\mathcal{S}_i$, group $\mathcal{G}_i$, or layer $\mathcal{L}_i$ as the $\beta_h$-percentile of the sorted hit heights,

within the $\beta_t$-percentage of the most recent hits in the slice, group, or layer. We adopt $\beta_h = 5\%$, and $\beta_t = 100\%$ by default, such that all hits within a given slice, group, or layer are used to derive the height of its base. It must be stressed that ampycloud does not apply any weighting to the different hits prior to computing the base height or sky coverage fraction of a specific slice,

---

[2]The recent release of an "airplane hit filter" by Vaisala may possibly warrant an adjustment of the default value of $\Theta_0$ in the future.

group, or layer. This choice is motivated by the following argument. ampycloud bundles ceilometer data over a certain time interval $\Delta t$ to obtain a (more) representative view of the sky. All cloud hits, irrespective of their age, are equally trusted in order to identify cloud layers. It thus makes sense to also treat them all equally when computing the cloud base height of the identified layers. For ampycloud, old hits are not less valid: they merely represent the state of the sky further away than the ceilometer lines-of-sight. Nonetheless, if a user were to prefer the cloud base heights to be more representative of the most recent hits within a slice/group/layer, the $\beta_t$ parameter can be set to lower values, e.g. $\beta_t = 30\%$. Note that changing the value of $\beta_t$ does not have any effect on the sky coverage fraction measurements.

As will be discussed in Sec. 2.6.3, the layering step is able to separate cloud layers very close from one another if they are well defined. The grouping step can also lead to sub-groups in very close proximity to each others when they are well defined, which can be problematic from a user perspective. The parameters $\Delta h_{l,\mathrm{vals}} = [250, 1000]$ and $\Delta h_{l,\mathrm{lims}} = [0, 10000, \infty]$ are introduced to ensure that groups and layers identified in the second and third step of the algorithm are sufficiently far apart from one another. Essentially, the base height of cloud groups/layers must be at least $\Delta h_{l,\mathrm{vals}}[k]$ ft apart if they have a base height in the range $[\Delta h_{l,\mathrm{lims}}[k]; \Delta h_{l,\mathrm{lims}}[k+1]]$ with $k \in [0, 1]$.

In the ampycloud Python package, two additional parameters $H_{\mathrm{MSA}}$ and $\Delta_{\mathrm{MSA}}$ are used to crop any hit with a height above aerodrome level significantly beyond the applicable Minimum Sector Altitude (MSA), namely beyond $H_{\mathrm{MSA}} + \Delta_{\mathrm{MSA}}$, where $H_{\mathrm{MSA}}$ is the MSA value (specified in ft aal, e.g. 10'000 ft aal and 8'000 ft aal for LSGG and LSZH, respectively) and $\Delta_{\mathrm{MSA}}$ is a buffer to properly treat cloud layers whose thickness/fluctuations extend slightly beyond the MSA.

## 2.5   The ampycloud diagnostic diagram

The ampycloud diagnostic diagram for a mock dataset, designed to demonstrate the role of each sequential step of the algorithm, in presented in Fig. 2. The details of each step will be discussed in Sec. 2.6.1 to 2.6.3. We do already note, however, that with a first robust *slicing* step being fine-tuned by the subsequent *grouping* and *layering* of cloud structures, the ampycloud algorithm is not overly sensitive to any of its parameter in particular.

Individual (artificial) cloud hits are visible in Fig. 2 as colored points in the diagram. This dataset simulates observations from 4 ceilometers, each performing (asynchronous) observations every 15 s, with a lookback time of 900 s. The color of each hit corresponds to the horizontal slice $\mathcal{S}_i$ (with $i \in [1, 2, 3, 4]$ in this example) they are assigned to by the first step of the algorithm. The colored rectangles expand $\pm \epsilon / 100 \cdot T(\mathcal{S}_i)$ above (below) the maximum (minimum) height of each slice $\mathcal{S}_i$, with $\epsilon = +10$ a parameter of ampycloud and $T(\mathcal{S}_i)$ the thickness of the slice $\mathcal{S}_i$. This vertical range is used to identify overlapping slices that may require grouping. Colored lines correspond to LOWESS fits of the cloud hit distribution within each slice: they are used to compute the slice fluffiness $f$ (in ft) used in the grouping step, that will be formally defined in Eq. 4. The black squares/triangles/circles around each cloud hit denote the final layer $\mathcal{L}_i$ (with $i \in [1, 2, 3, 4]$ in this example) they are assigned

**Table 2.** Parameters (and associated default values) controlling the behavior of ampycloud as of version 2.0.0 of the eponymous Python package. The values of each parameter were chosen and verified using specific examples and a large-scale statistical assessment of the algorithm's accuracy (see Sec. 3.4). The meaning and role of each parameter are described in detail in the associated article sections. Note that the parameter specifying the aerodrome's MSA must be specified in ft aal (and not in ft amsl).

| Parameter | Value | Unit | Python variable name |
|---|---|---|---|
| General — Sec. 2.4 | | | |
| $H_{\mathrm{MSA}}$ | LSGG: 1e4, LSZH: 8e3 | ft aal | `MSA` |
| $\Delta_{\mathrm{MSA}}$ | 1500 | ft | `MSA_HIT_BUFFER` |
| $\Theta_0$ | 3 | - | `MAX_HITS_OKTA0` |
| $\Theta_8$ | 1 | - | `MAX_HOLES_OKTA8` |
| $\beta_h$ | 5 | % | `BASE_LVL_HEIGHT_PERC` |
| $\beta_t$ | 100 | % | `BASE_LVL_LOOKBACK_PERC` |
| $\Delta h_{l,\mathrm{vals}}$ | [250, 1000] | ft | `MIN_SEP_VALS` |
| $\Delta h_{l,\mathrm{lims}}$ | $[0, 10000, \infty]$ | ft | `MIN_SEP_LIMS` |
| Slicing step — Sec. 2.6.1 | | | |
| $\alpha_s$ | 0.2 | - | `SLICING_PRMS['distance_threshold']` |
| $\tau_s$ | $10^5$ | s | `SLICING_PRMS['dt_scale']` |
| $\Delta h_{s,\mathrm{min}}$ | 1000 | ft | `SLICING_PRMS['height_scale_kwargs']['min_range']` |
| Grouping step — Sec. 2.6.2 | | | |
| $\epsilon$ | +10 | % | `GROUPING_PRMS['height_pad_perc']` |
| $\tau_g$ | 180 | s | `GROUPING_PRMS['dt_scale']` |
| $\Delta h_{g,\mathrm{min}}$ | 100 | ft | `GROUPING_PRMS['height_scale_range'][0]` |
| $\Delta h_{g,\mathrm{max}}$ | 500 | ft | `GROUPING_PRMS['height_scale_range'][1]` |
| $L_{\mathrm{frac}}$ | 0.35 | - | `LOWESS['frac']` |
| $L_{\mathrm{it}}$ | 3 | - | `LOWESS['it']` |
| Layering step — Sec. 2.6.3 | | | |
| $\Theta_{\mathrm{min}}$ | 2 | okta | `LAYERING_PRMS['min_okta_to_split']` |
| $\delta$ | 0.95 | - | `LAYERING_PRMS['gmm_kwargs']['delta_mul_gain']` |
| $\gamma$ | 100 | - | `LAYERING_PRMS['gmm_kwargs']['rescale_0to']` |

to, as a result of the third step of the algorithm. The intermediate groups $\mathcal{G}_i$ (with $i \in [1,2,3]$ in this example) identified by the second algorithm step are not shown for readability purposes.

Secondary x/y axes on the right/top of the diagram show the rescaled hit times/heights axis used for the slicing and grouping steps. The characteristics of each slice $\mathcal{S}_i$/group $\mathcal{G}_i$/layer $\mathcal{L}_i$ with a sky coverage fraction of FEW or more are shown on the right-hand-side of the diagram, including the measured sky coverage in oktas. Overlapping slices are tagged with ⇌. Groups for which the layering step was executed are tagged with $-$, $=$, or $\equiv$ depending if 1, 2, or 3 distinct sub-layers were identified. Cloud slices/groups/layers found are reported according to METAR syntax, i.e. [FEW|SCT|BKN|OVC]$nnn$, where $nnn$ is the

cloud slice/group/layer height in hundreds of feet above aerodrome level. Not all cloud slices/groups/layers need to be reported according to the ICAO rules (ICAO, 2018). The relevant ones are boxed.

The final selection of cloud layers is shown in the top right of the Figure. To the bottom left of the diagram, the number of ceilometers contributing data to the diagram is indicated explicitly, alongside the maximum number of cloud hits possible

per cloud layer (240 in this example) corresponding to the total of the individual time steps reported by each ceilometer. The sup-/subscripts indicate the values of the parameters $\Theta_8$ and $\Theta_0$ (with the format 8:$\Theta_8$ and 0:$\Theta_0$). One should note that the number of slices, groups, and layers reported on the right-hand-side of the diagrams do include clusters that are comprised of less than $\Theta_0$ hits, as illustrated by the single cloud hit at (-800 s; 3100 ft aal) that gets assigned its own slice/group/layer as indicated by the dotted horizontal line in the diagram.

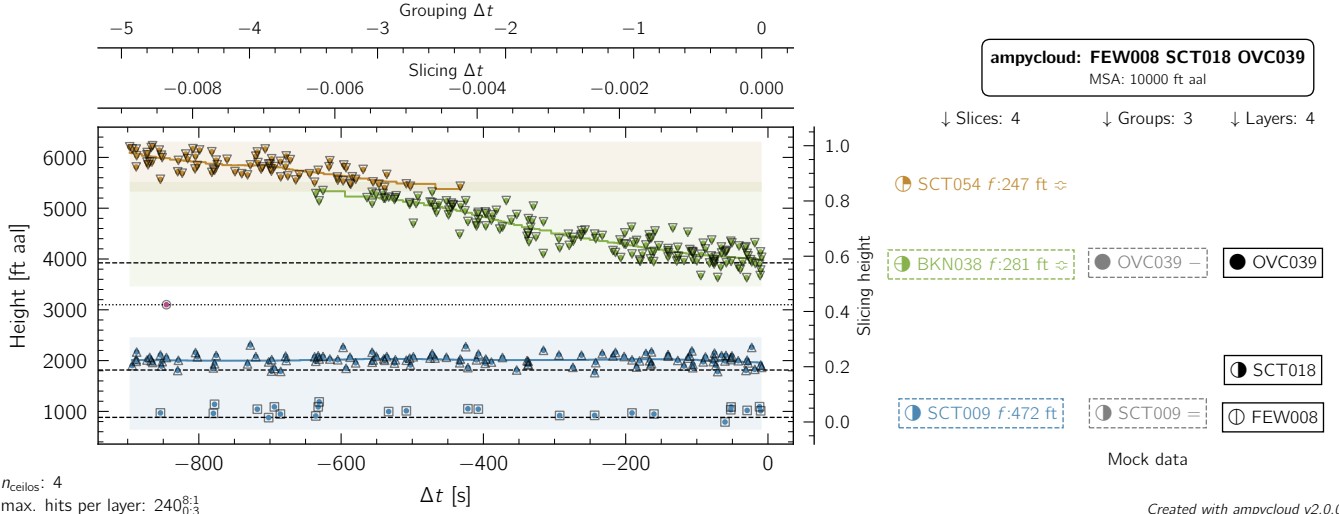

**Figure 2.** ampycloud diagnostic diagram for a mock (demonstration) dataset with a 900 s lookback time, designed to illustrate the slicing, grouping, and layering steps of the algorithm. See Sec. 2.5 for a detailed description of the different elements of this Figure.

 ## 2.6   The three ampycloud steps

### 2.6.1   Slicing

Given a set of hits gathered over a time interval $\Delta t$, the first ampycloud step consists in the identification of (horizontal) height "slices" $\mathcal{S}_i$. The algorithm does so using an agglomerative clustering approach with an average linkage criterion, and a Manhattan metric[3]. The key to the robustness of this processing step lies in the rescaling of the cloud base hits, both along the time 250 and height dimensions, applied prior to the clustering.

Given the minimum and maximum hit heights $h_{\min}$ and $h_{\max}$, the interval $[h_{\min}; h_{\max}]$ is mapped linearly onto the interval $[0; 1]$. If, however, $h_{\max} - h_{\min} < \Delta h_{s,\min}$, it is the interval:

$$[\frac{h_{\max} + h_{\min}}{2} - 0.5\Delta h_{s,\min}; \frac{h_{\max} + h_{\min}}{2} + 0.5\Delta h_{s,\min}]$$

with length $\Delta h_{s,\min}$ that gets mapped onto $[0; 1]$ instead. Essentially, $\Delta h_{s,\min}$ (which is a parameter of ampycloud, see Table 2 for the complete list) ensures that cases with a small height dispersion do not get over-stretched, and thus "over-sliced".

Along the time axis, hits are rescaled (i.e. normalized) by a very large number $\tau_s$. If $\tau_s$ is large enough, the distance between hits along the time direction becomes essentially negligible in the rescaled space. As a result, the measure of the linkage 260 distance –used by the agglomerative clustering method to decide whether to merge hits or not– is dominated by the height component of the hits[4]. In this rescaled space, a distance threshold $\alpha_s$ will identify horizontal slices of hits whose rescaled mean heights are separated by no more than $\alpha_s$. With $\tau_s \gg 1$, the value of $\alpha_s$ represents the maximum thickness of individual horizontal slices, expressed as a fraction of the total height range in the sample. Hence, at most $1/\alpha_s$ distinct slices can be identified by ampycloud, and these will never be thinner than $\alpha_s \cdot \Delta h_{s,\min}$. For the default ampycloud parameters, $1/\alpha_s = 5$ 265 slices and $\alpha_s \cdot \Delta h_{s,\min} = 200\,\mathrm{ft}$.

This rescaling scheme is well suited to identify vertically-stratified cloud layers with a stable base as a function of time. However, it is ill-suited for cloud layers with a thick and/or time varying base height, where the variation is larger than $\alpha_s \cdot (h_{\max} - h_{\min})$. This shortcoming is illustrated in Fig. 2 with the top layer being *sliced* in two. The slicing step is also 270 ill-suited to distinguish isolated cloud layers separated by less than $\alpha_s \cdot (h_{\max} - h_{\min})$. This is illustrated with the bottom two cloud layers in Fig. 2, that are detected as a single slice.

The grouping and layering steps of ampycloud are designed specifically to handle these limitations of the slicing step. As a result, the output of ampycloud is not very sensitive to the values of $\Delta h_{s,\min}$ and $\alpha_s$, in that any over-/under-slicing of hits is 275 compensated by the subsequent processing steps.

---

[3]implemented in the Python package via the sklearn package (Pedregosa et al., 2011).

[4]The attentive reader may wonder why the slicing step relies on a 2-D clustering algorithm, given that the use of $\tau_s \gg 1$ essentially implies that 1-D clustering approach would be equivalent. The reason for this is historical, and related to the initial assembly of the ampycloud algorithm.

### 2.6.2 Grouping

This processing step is designed to handle the *over-slicing* of coherent cloud hit structures as a result of a broad height span or marked up-/downward trends of a cloud layer (as illustrated by the top two slices in Fig. 2). It begins by identifying so-called *overlapping* slices. These are flagged by the symbol "↷" in the ampycloud diagnostic diagrams. By (our) definition, slices $i$ and $j$ overlap if one of the following two conditions is met:

$$\min(\mathcal{S}_i) - \frac{\epsilon}{100} \cdot T(\mathcal{S}_i) < \max(\mathcal{S}_j) + \frac{\epsilon}{100} \cdot T(\mathcal{S}_i) \tag{2}$$

$$\max(\mathcal{S}_i) + \frac{\epsilon}{100} \cdot T(\mathcal{S}_i) > \min(\mathcal{S}_j) - \frac{\epsilon}{100} \cdot T(\mathcal{S}_i) \tag{3}$$

with $\min(\mathcal{S}_i)$ and $\max(\mathcal{S}_i)$ the minimum and maximum hit heights of the slice $\mathcal{S}_i$, $T(\mathcal{S}_i) = \max(\mathcal{S}_i) - \min(\mathcal{S}_i)$ the thickness of the slice $\mathcal{S}_i$, and $\epsilon$ a parameter of the ampycloud algorithm. Setting $\epsilon = 0$ would result in a strict overlap diagnostic. To avoid edge cases with identical minima and maxima cloud hit heights, as well as to account for natural cloud base height fluctuations, we adopt a default value of $\epsilon = 10\%$ to slightly *pad* the slices by a fraction of their thickness before checking for overlap. For example, the top two slices in Fig. 2 (green and red) overlap according to this criterion with our adopted value of $\epsilon = +10\%$.

Having identified a set of $n$ overlapping slices, ampycloud uses an agglomerative clustering method with a single linkage criterion and an Euclidean metric to identify coherent sub-groups $g_k$, with $k \in [1; 2; \ldots]$. Each sub-group $g_k$ is subsequently assigned to a master group $\mathcal{G}_i$ based on the slice $\mathcal{S}_i$ to which the majority of the hits in group $g_k$ belong. Hence, the ampycloud grouping of $n$ overlapping slices will result in no less than 1 and no more than $n$ master groups $\mathcal{G}$. In other words, there can be no more groups $\mathcal{G}$ than slices $\mathcal{S}$. Essentially, the slicing step of ampycloud is constraining the grouping step, which would otherwise be overly sensitive to noisy datasets. It also ensures that sparse cloud layers do not get broken into sub-groups along the time axis.

The hit heights and (relative) times are being rescaled prior to being inputted to the (single linkage) agglomerative clustering method of the grouping step. The rescaling is performed separately for each set of overlapping slices. In all cases, the time axis scaling factor is set to $\tau_g$. The height scaling factor is taken to be the minimum slice fluffiness $f_{\min}$ in the overlapping bundle, provided that $f_{\min} \in [\Delta h_{g,\min}, \Delta h_{g,\max}]$. If $f_{\min}$ is larger (smaller) than $\Delta h_{g,\max}$ ($\Delta h_{g,\min}$), the latter value is used as the height rescaling factor instead. The *fluffiness* $f$ of a slice $\mathcal{S}_i$ is a measure of the spread of cloud hits around the smoothed cloud height trend, which we formally define as:

$$f(\mathcal{S}_i) = 2 \cdot \frac{1}{N} \sum_{h_k \in \mathcal{S}_i} |h_{\mathrm{hgt},k} - L(h_{\mathrm{tme},k})| \tag{4}$$

with $N$ the number of cloud hits in the slice, $h_{\mathrm{hgt},k}$ the individual cloud hit heights, and $L(h_{\mathrm{tme},k})$ the LOWESS-derived (Cleveland, 1979) height corresponding to the hit time $h_{\mathrm{tme},k}$. The LOWESS[5] algorithm[6] is used to robustly measure the mean

---

[5]Locally Weighted Scatterplot Smoothing

[6]implemented in the ampycloud Python package via the statsmodels package (Seabold and Perktold, 2010).

cloud hit height as a function of time using a series of localized weighted linear regressions. The LOWESS fit of each slice in Fig. 2 is shown using colored lines. It relies on two parameters: the fraction of slice hits (along the time axis) $L_{\mathrm{frac}}$ to use for each linear regression, and the number of iterations $L_{\mathrm{it}}$ for each fit. The LOWESS fit of a given slice $\mathcal{S}_i$ is fully sensitive to coherent cloud hit height fluctuations affecting more than a fraction $L_{\mathrm{frac}}$ of the slice's cloud hits. The smaller below $L_{\mathrm{frac}}$ the fractional duration of a specific height variation, the smaller is its influence on the resulting LOWESS fit. By default, we adopt $L_{\mathrm{frac}} = 0.35$ which corresponds to a "sensitivity timescale" of $\sim$5 min for a uniform cloud hit distribution observed over an interval of 15 min. The default value of $L_{\mathrm{it}} = 3$ ensures that the final LOWESS fit is not affected by spurious hits with significant height offsets. We note that although the ampycloud algorithm uses the fluffiness of (overlapping) slices $\mathcal{S}$ only for the grouping step, the ampycloud Python package computes the fluffiness of every slice $\mathcal{S}_i$, group $\mathcal{G}_i$, and layer $\mathcal{L}_i$ by default.

The agglomerative clustering method with single linkage criterion uses the closest elements of sub-clusters to decide whether to merge them or not. With a linkage distance of 1 fixed inside ampycloud, $\tau_g$ and the slice fluffiness determine the maximum distance (in time and height, respectively) below which two cloud hits will be assumed to belong to the same cloud structure. Unlike the slicing step that ignores the time information (with $\tau_s \gg 1$), the grouping step is much better suited to track coherent height variations as a function of time, as demonstrated in Fig 2. In that example, the hits in the top two slices (red and green at 3'800 ft and 5'400 ft, respectively) are grouped into a single master group with a base at 3'800 ft.

### 2.6.3 Layering

By design and as illustrated in Fig. 2, the slicing step will inevitably bundle distinct cloud structures into common slices if isolated cloud layers are separated by less than $\alpha_s \cdot (h_{\max} - h_{\min})$. The layering step is designed to correct this shortcoming. It relies on the computation of Gaussian mixture models with 1, 2, and 3 components for every master group $\mathcal{G}_i$ with a sky coverage fraction of at least $\Theta_{\min}$ oktas. For each model, the value of the associated Bayesian Information Criterion $BIC_i$ is recorded.

Statistically speaking, the best fitting model out of the three will have the lowest $BIC$ score. However, cloud hits belonging to a given cloud layer typically will not follow a Gaussian distribution. Using the relative likelihood (and associated probabilities) of different Gaussian Mixture models to identify the most likely number of components in a group would thus typically lead to an overestimation: not because more components represent a better fit, but rather because they represent a somewhat-less-but-still-terrible one (see Appendix A for details). Hence, ampycloud does not simply follow this criterion alone to decide whether a given group requires to be broken up. The following selection approach is adopted instead.

The starting assumption for each group $\mathcal{G}_i$ is that it does not contain sub-layers. Moving sequentially through the Gaussian Mixture models with 1, 2, and 3 components, ampycloud will identify $n$ sub-layers if $BIC_n < \delta \cdot BIC_{\mathrm{current\ best}}$. Essentially, ampycloud assumes $n$ sub-layers to be present only if the decrease in the model's $BIC$ score over the current best amount of

sub-layers is greater than $(1 - \delta)$.

$BIC$ scores are sensitive to the data values in the underlying dataset. To alleviate the consequences of this fact, ampycloud rescales the height range of every individual master group $\mathcal{G}_i$ between 0 and $\gamma$ prior to searching for sub-layers. A given $\delta$ value will thus split groups based upon the relative separation of their sub-layers in terms of their height dispersion, irrespective of

whether the groups are located at 2000 ft or 20'000 ft.

This approach still remains unstable for small datasets. This is why ampycloud will not attempt to break-up groups with a sky coverage fraction of less than $\Theta_{\min} = 2$ oktas. The adopted values of $\delta = 0.95$ and $\gamma = 100$ imply that two perfectly Gaussian sub-layers would be separated by ampycloud if their means are offset by at least $\sim$5 standard deviations from one

another (see Appendix A for details).

## 3 The ampycloud Python package

### 3.1 Implementation

The ampycloud Python package (Vogt and Regenass, 2024) is developed on GitHub (ampycloud, 2024b) as a regular, non-

MeteoSwiss specific, pip-installable Python package. The source code is made available to the general community as open-source software under the terms of the 3-Clause BSD License. As a project, ampycloud follows a Continuous Integration/-Continuous Delivery approach. It relies on GitHub Actions to run automated quality/stability/validity checks, publish the documentation, and upload new versions to the Python Package Index (PyPI). The algorithm described in this article corresponds to version 2.0.0 of the ampycloud Python package. The scientific behavior of this version is essentially identical to

version 1.0.0 that was deployed at LSGG in November 2023. Version 2.0.0 simply provides a more coherent variable naming scheme, and an improved handling of NSC/NCD codes, in addition to a few minor code updates.

As a key dependency of the operational AUTO METAR software of MeteoSwiss, ampycloud must remain robust and stable. An evolution of the code over time is inevitable and even desirable, but it will be carefully controlled. Any future release of the

code (and associated changes) will be fully documented in the online documentation of the ampycloud package (ampycloud, 2024a), to which we refer the interested readers for more details on the evolution of the algorithm after the publication of this article. We shall not discuss here the technical implementation of the ampycloud Python package itself in extensive detail. It suffices to say that 1) ampycloud requires cloud and VV hits stored in a suitably-formatted pandas `DataFrame` as input, 2) the slicing and grouping steps rely on the `sklearn.cluster.AgglomerativeClustering()` class from

the scikit-learn Python package, 3) the layering step relies on the `sklearn.mixture.GaussianMixture()` class from the same package, and 4) the fluffiness of slices/groups/layers is computed using the LOWESS algorithm implemented in the

statsmodels package. We refer the interested readers to the official ampycloud online documentation (ampycloud, 2024a) for further information on its different classes and functions.

## 3.2 Deployment at MeteoSwiss

The ampycloud algorithm and associated Python package has been used at LSGG since 2022-10-31 outside the operational hours of the aerodrome, with its outcome supervised by an AMO during operational hours and used as METAR Proposals. The initial version 0.5.0 was replaced by version 1.0.0 of the code on 2023-11-28. The version 2.0.0 of the code, developed as part of the publication of this article, will be deployed in the near future. As of 2024-05-01, ampycloud operates (alongside SM△RT) in standalone mode (i.e. without human interaction) at LSGG also during the aerodrome's operational hours.


In the MeteoSwis operational setup, the ampycloud Python package is imported as an external dependency within a Python service known as the *cloud calculator* (closed-source software). The cloud calculator is one of several calculators that comprise the autometpy software, developed as part of the AMAROC program. The cloud calculator runs on the MeteoSwiss OpenShift container platform, where it stands ready to receive calculation requests. The requests are issued and orchestrated by SM△RT, 385 that decides how often and with what data the cloud calculator is called. At LSGG, SM△RT makes 3 requests to the cloud calculator every minute in:

- AUTO METAR mode, using data from all 4 ceilometers over the last 15 min.

- AUTO MET REPORT mode using data from the 2 ceilometers representative of RWY22 over the last 6 min.

- AUTO MET REPORT mode using data from the 2 ceilometers representative of RWY04 over the last 6 min.

Monitoring the status and computational performances of the cloud calculator is done using a flexible dashboard providing access to various metrics (CPU load, memory use, request time, number of requests, errors). Dedicated log files are also collected and can be queried whenever necessary. Diagnostic plots are produced by a separate calculation request using the same input data for monitoring and debugging purposes, in particular to quickly answer questions from stakeholders.

## 3.3 Computational speed (performance)

We use the mock dataset presented in Fig. 2 to keep track of the speed performance of the ampycloud package over the course of its development, by means of a dedicated GitHub Action (ampycloud, 2024c). With ampycloud v2.0.0, the processing of this dataset takes 0.20 s on average on a 16-inch MacBook Pro 2019 with an 2.3 GHz 8-Core Intel Core i9. On a server running Ubuntu with Linux kernel 6.2 with 4 cores, the processing lasts 0.30 s (ampycloud, 2024d). These values are representative of ampycloud processing times in general. Among the real examples presented in Appendix B, the case in Fig. B2 takes the most 400 time to process (0.35 s), whereas the case in Fig. B12 takes the least time (0.12 s). These processing times do not include the creation of the ampycloud diagnostic diagrams, which are optional and require an additional $\sim 2$ s per diagram.

### 3.4 Comparison with reference METARs (accuracy)

We present in Appendix B a series of representative, real examples from the Swiss civil aerodromes. These are meant to il-
lustrate the behavior of the ampycloud algorithm under varying weather conditions. For each case, the actual METAR cloud
codes (produced by an AMO at the time) are provided for comparison purposes. To maximize the readability of this article,
each case is discussed directly within the figure caption.

Comparing AUTO METAR cloud codes (derived purely from ceilometer observations) with METAR ones (derived fully or
in part from AMOs) is evidently subject to a series of fundamental caveats. Both methods have specific strengths and limi-
tations (see e.g. paragraph 7.4.3.1 in ICAO, 2011). For example, cloud layers in the 1-2 okta range are systematically more
difficult to observe at night for human observers, which is not the case for ceilometers (Boers et al., 2010). On the other hand,
the exact ceilometer performances can vary depending on air temperature and direct sun exposure. Standard ceilometers typi-
cally only sample a small part of the sky immediately above them, whereas AMOs can more easily observe the entire sky (even
though the slanted view from a fixed position still limits their ability to detect holes in cloud layers with high sky coverage
fractions). Ceilometers will typically require to collect data over a longer time frame than AMOs to derive the appropriate cloud
codes, such that METARs cloud codes will typically be more representative of the "instantaneous" situation at the message
distribution time, when compared to AUTO METARs. Finally, AMOs at Swiss aerodromes (at least) will typically estimate
sky coverage fractions independently, but remain encouraged to use ceilometer data for their estimation of cloud base heights.
The extent to which they do so depends on the weather conditions and the personal inclination of each observer.

Bearing these differences in mind, the overall accuracy of the ampycloud algorithm was evaluated in a statistical sense, using
a reference dataset of 2128 cases extracted over a 5-year period (2018-2022) from LSGG METARs. Doing such a comparison
for LSGG remains meaningful and important: not because METARs represent the absolute truth, but rather because AMOs
were the existing operational system. Any significant/systematic change in the behavior of AUTO METARs with respect to
METARs must be characterized, documented, and explained to the relevant Civil Aviation Authority and stakeholders (e.g. air
traffic controllers). The 2128 reference cases were selected to provide a representative sub-sample of all the METARs over this
period. The case selection focused on operationally-relevant situations (according to the METAR, ampycloud, or both). Of
the 2128 cases, 71.5% correspond to a cloud "ceiling" situation, i.e. the presence of a BKN or OVC cloud layer. Without any
pre-selection, the ratio of ceiling cases at LSGG is ∼36%.

Using the sample of 2128 cases, we present in Fig. 3 a comparison of the cloud layers with the highest sky coverage fraction
as identified by the ampycloud algorithm and the actual METAR. In $57.8\%$ of the cases, the ampycloud output is consistent
with the METAR, and at most 1 category off in $95.7\%$ of the cases. The cases where ampycloud differs more significantly (at
least 2 classes) from the METAR land in two categories. The cases when ampycloud underestimates the sky coverage fraction
of the cloud layer (aka "misses"; $3.2\%$ of the cases) are related to situations where cloud layers are not seen by the ceilometers

- either because they are not above the line of sight, or because the ceilometer signal is blocked by a lower layer. This also explains the 6.3% of misses where ampycloud returns NCD/NSC and the AMO reports FEW: a situation typically related to stationary cumulus clouds above the Jura mountain range, at a distance of ~12 km from the aerodrome. The "false alarms" (1.2% of all cases), where ampycloud overestimates the sky coverage fraction of the cloud layer by at least 2 classes, are nearly all related to cases of low-level fog seen as a low-level cloud layer by the ceilometers. For operations, these cases are treated by a separate vertical visibility algorithm, which also uses horizontal visibility and present weather information (i.e. the presence of snow and fog).

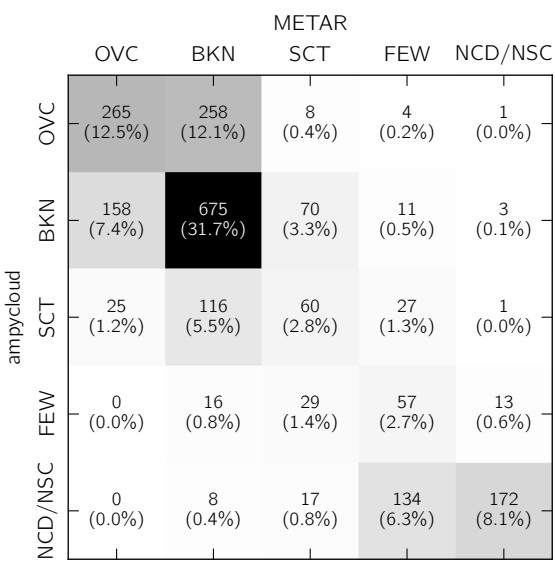

**Figure 3.** Matrix comparing the categorization of the cloud layers with the highest sky coverage fraction identified by the ampycloud algorithm (vertical axis) with the actual METAR validated by a AMO (horizontal axis), for a reference dataset of 2128 cases assembled from LSGG observations. Boxes located above the diagonal can be associated to "false alarms" where ampycloud finds a sky coverage fraction larger than the METAR. The cases below the diagonal can be understood as "misses", where the opposite happens.

The same information can be also be presented using operationally-relevant categories. In Fig. 4, we present the ability of ampycloud to report a cloud ceiling. The ampycloud identification of a ceiling (or absence thereof) is in agreement with the corresponding METAR 87.7% of the time. The comparison of Fig. 3 and Fig. 4 shows that ceiling false alarms (4.6%) and misses (7.8%) are dominated (respectively) by cases where ampycloud characterizes a SCT layer as BKN (3.3% of the cases), and a BKN layer as SCT (5.5% of the cases). These two types of situations are a direct consequence of the limited spatial sampling capabilities of ceilometers. Of course, one cannot rule out the possibility that some of these mismatches are driven by the limited ability of human observers to distinguish between cloud layers with a sky coverage fraction of slightly more or

less than 56.25% (which corresponds to the limit between SCT and BKN, see Boers et al., 2010).

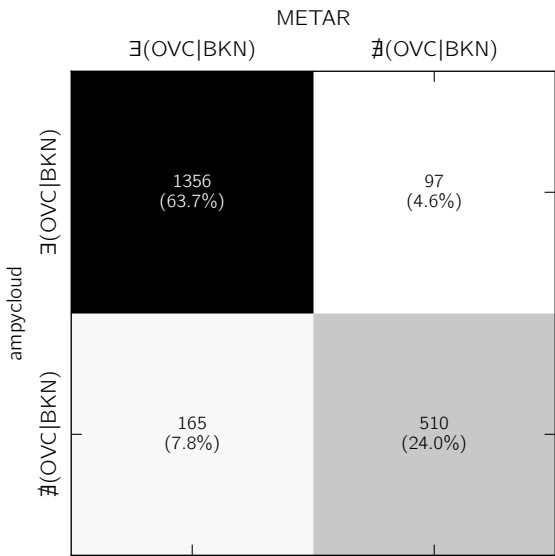

**Figure 4.** Same as Fig. 3, but with simplified categories related to the presence/absence of a cloud ceiling.

The matrix in Fig. 3 is useful to identify cases with strong deviations in the sky coverage fraction. However, detecting the
correct sky coverage fraction for a given cloud layer does not guarantee that the cloud layer is detected at the correct height.
We thus provide in Fig. 5 a comparison of the height of cloud layers with the highest sky coverage fraction measured by
ampycloud with respect to that reported in the corresponding METAR. In $64.4\%$ of the cases, ampycloud identifies a cloud
layer height in good agreement with the METAR. The majority of the remaining cases are comprised of situations when
ampycloud provides somewhat lower cloud base heights than the METARs. This is a direct consequence of the conservative
approach of the ampycloud algorithm, that considers the entire set of cloud and VV hits (in its default configuration) to derive
a given cloud base height. The example in Fig. B11 illustrates this behavior. Cases in the bottom row (6.6% of the total) cor-
respond to situations where the cloud layer is being missed entirely by the ceilometers over the duration of the time interval $\Delta t$.

## 4 Limitations and future prospects

The ampycloud Python package focuses tightly on the implementation of the ampycloud algorithm itself, with only a handful
of dependencies, all well-maintained and under active development within the Python community. The ampycloud package is
not designed to interact with the outside World (and other programming languages, e.g. via JSON format). The implementation
of the necessary Application Programming Interfaces (APIs) are left to the interested users. ampycloud is also not designed

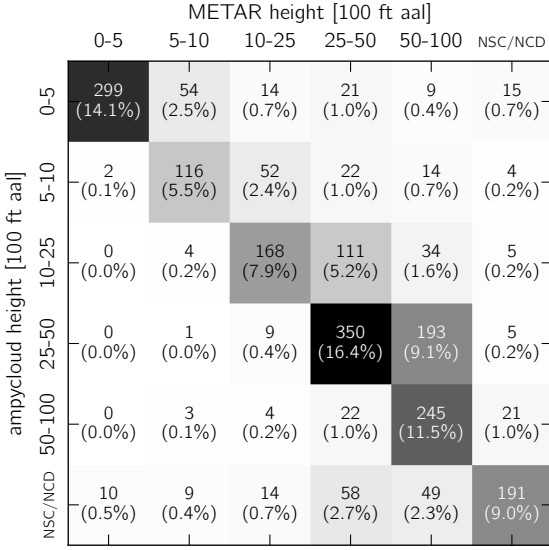

**Figure 5.** Same as Fig. 3, but for the height of the densest cloud layer. The differences between ampycloud and the METARs are dominated by cases where ampycloud returns a cloud layer height somewhat lower than the METAR. This is a direct consequence of the conservative design of the algorithm, that (in its default configuration) considers the entire look-back time to compute the height of a given set of cloud and VV hits.

to handle partial/complete ceilometer failures or data transmission issues: it will simply process the cloud and VV hits that one feeds it. From this perspective, the handling of operational exceptions and errors must be done by the user before calling ampycloud.

In its current implementation, ampycloud uses a fixed time interval $\Delta t$ to characterize cloud layers. In doing so, ampycloud is essentially trading-off its responsiveness to rapidly evolving conditions with its ability to obtain a more representative view of the entire sky. The default time interval $\Delta t = 15$ min adopted by MeteoSwiss is shorter than in other algorithms with similar goals (see e.g. Nadolski, 1998; Wauben, 2002; ICAO, 2011). But unlike those algorithms that give additional weight to the most recent measurements, ampycloud treats every hit equally. This implies that ampycloud may possibly respond somewhat less rapidly to changes, but should be more stable when analyzing sparse cloud layers. We note that increasing the value of $\Delta t$ is not necessarily a guarantee for a better match with the AMOs. At LSGG for example, there are often cumulus (humilis and/or mediocris, which are not operationally relevant) that remain stationary over the Jura mountain ridge, and that can never be detected by the aerodrome's ceilometers. Several cases presented in Appendix B have the lowest cloud layer missed entirely by the ceilometers (see Figs. B4, B6, B7, B8, and B10). Had one used a time interval $\Delta t = 30$ min instead of 15 min, the lowest cloud layers would still have been missed entirely by the ceilometers in each of these cases. Understanding the exact behavioral differences between ampycloud and other algorithms with similar purpose will require a dedicated comparison of

485 their respective accuracy against a reference set of ceilometer data (either real, or simulated), which is outside the scope of this article. Up to now, such comparisons were essentially impossible due to the private nature of source codes and lack of detailed documentation. With this article and its accompanying material (Vogt, 2024), we purposely seek to make the ampycloud processing of the examples shown in Figs. B1 to B12 reproducible by motivated users and researchers.

490 It must be stressed that ampycloud does not challenge the quality/nature of the cloud and VV hits that it is being provided: it trusts them all fully and equally. The capacity of the algorithm to provide an accurate assessment of cloud layers above an aerodrome is thus directly limited by the ability of ceilometers to report clouds up to the aerodrome's MSA in the first place, and measure their heights accurately (Costa-Surós et al., 2013; Wagner and Kleiss, 2016; Kotthaus et al., 2016; Illingworth et al., 2019). This (current) reliance of ampycloud on cloud base hits (as reported by ceilometers via black-box algorithms) 495 should not be overlooked by users interested to combine datasets from multiple ceilometer brands/models. Any spurious cloud hit (be it caused by measurement noise, rain, or airplanes flying overhead) will also inevitably impact the outcome of ampycloud. So is a "lack of cloud hits", for example when the view towards upper layers is blocked by lower ones in complex situations. ampycloud does not do any upward correction of the cloud coverage of the upper layers (yet), e.g as done in the ASOS algorithm (see e.g. Appendix A, Sect. Cloud layers (8), Example 1 in ICAO, 2011). Another enhancement possibility 500 for the ampycloud algorithm would be to use the wind speed to identify suitable lookback times as a function of height, in order to obtain a (more) representative sampling of the sky. Doing so in real-time could be achieved, at least up to the aerodrome minimum sector altitudes, by direct measurements (for example using wind profilers), by data extraction from numerical models, or by a combination of both.

505 The slicing step of ampycloud relies on the rescaling factor $\tau_s$ to reduce the dimensionality of the 2D clustering to almost 1-D. A further development might be to use a robust 1-D clustering method instead. The layering step of ampycloud is the part of the algorithm that shows the most potential for improvement, for the following reasons. Unlike the slicing and grouping steps, the parameters $\gamma$ and $\delta$ associated to this step are less-directly relatable to physical quantities. For specific distributions of cloud hits, this step can be sensitive to the random seed used by the system[7]. Most importantly, the use of Gaussian Mixture 510 Models clearly is not ideal from a physics perspective, given the fact that cloud base hits are not expected (nor seen) to systematically follow a Gaussian distribution, although it remains a sufficiently valid approximation.

The main limitation of ampycloud, however, currently resides in the fact that it relies exclusively on cloud and VV hits, that are processed without questions. The use of complete backscatter profiles from the ceilometers could help improve this 515 state-of-affair: for example, by enabling the identification of spurious hits triggered by heavy precipitation or aircraft, and/or by supplementing cloud base height measurements with cloud layer thickness which could significantly ease the identification of coherent structures. The use of complete backscatter profiles could also enable the identification of the maximum observed height (in case of thick cloud layers), and better account for the sky coverage fraction of partially-obscured cloud

---

[7]This seed is fixed by ampycloud to ensure that results are strictly repeatable on a given system.

layers, while removing any reliance on black box algorithms (developed by ceilometer manufacturers to detect cloud base hits).

The use of a finite number of ceilometers alone cannot always be sufficient to differentiate very sparse cloud layers from a "true" clear sky. If a series of ceilometers detect no clouds, the use of pyrgeometers (see e.g. Aviolat et al., 1998; Marty and Philipona, 2000; Dürr and Philipona, 2004), visible all-sky cameras (Wacker et al., 2015), infrared all-sky imagers (Aebi et al., 2018) or even satellite images could help ascertain the validity of a sky-clear condition – albeit without the ability to estimate the height of the sparse cloud layers that may have been missed by the ceilometers with the same accuracy. Similarly to Doppler lidars, the use of rotating beam ceilometers (WMO, 2021) with the ability to probe more than one sightline could clearly improve the reliability of AUTO METARs for sparse cloud layers (i.e. for intermediate okta values), albeit at an increased financial cost and maintenance challenge (a rotating mechanical component is more prone to technical issues).

## 5 Conclusions

The ampycloud algorithm was developed at MeteoSwiss as part of a large effort to fully automate the production of METARs at Swiss civil aerodromes. Its specific prupose is to determine the sky coverage fraction and base height of cloud layers using ceilometer data (in the form of individual cloud base hits). The eponymous Python package is released online as open-source software under the terms of the 3-Clause BSD license. The ampycloud Python package forms an integral part of the software infrastructure of MeteoSwiss. Yet, it does not contain any MeteoSwiss-specific code nor does it rely on any specific hardware/-software infrastructure (which are all contained within the autometpy software, see Sec. 3.2). The code and its dedicated online (technical) documentation is completed by this article describing the underlying algorithm and its scientific motivation.

The accuracy of the ampycloud algorithm has been tested in detail on a series of specific examples (as illustrated in Figs. B1 to B12), and statistically over a reference set of cases extracted over a 5-year period. With a correct identification of a ceiling (or absence thereof) 87.7% of the time, the results of the ampycloud algorithm are found to be in good agreement with the corresponding METARs, such that the use of this algorithm at LSGG was approved by the Swiss Civil Aviation Authority.

The automatic production and dissemination of AUTO METARs without human supervision at LSGG started on 2024-05-01. Nonetheless, several elements of the ampycloud algorithm have potential for further improvement. The necessity and exact benefits of these improvements will be continuously investigated and evaluated by MeteoSwiss over the coming years, throughout and beyond the transition from METAR to AUTO METAR at Swiss civil aerodromes.

The public release of ampycloud has taken place within a large paradigm change towards Open Government Data in Switzerland (Assemblée fédérale de la Confédération suisse, 2023). An open-source software evidently facilitates additional testing of the algorithm at various locations beyond the Swiss civil aerodromes. Most importantly, we hope that it will ease the implementation of dedicated intercomparison campaigns to evaluate the accuracy of the various cloud algorithms deployed at

aerodromes worldwide.

*Code and data availability.* The ampycloud Python package is freely available on GitHub (https://github.com/MeteoSwiss/ampycloud), with each release archived on Zenodo (DOI:10.5281/zenodo.8399683). The script and cloud hits used to generate the Figures in Appendix B have also been stored on Zenodo (DOI:10.5281/zenodo.10171151), from where they can be downloaded freely.

## Appendix A: The ampycloud Gaussian mixture model selection rule

In ampycloud, Gaussian mixture models with 1, 2 and 3 components are used to determine whether a given group $\mathcal{G}$ is comprised of multiple distinct sub-layers. To illustrate how the Bayesian Information Criterion ($BIC$) scores allow us to reliably select the optimal number of sub-layers in a given group, we create a series of artificial distributions of cloud hits. Each distribution is comprised of either 2 or 3 Gaussian layers, each with a number of randomly-generated hits $N_{\text{hits}} \in [10, \ldots, 350]$, and with the layers separated in height by up to $\Delta = 8$ times their standard deviation $\sigma$. Four examples of these distributions are presented in Fig. A1.

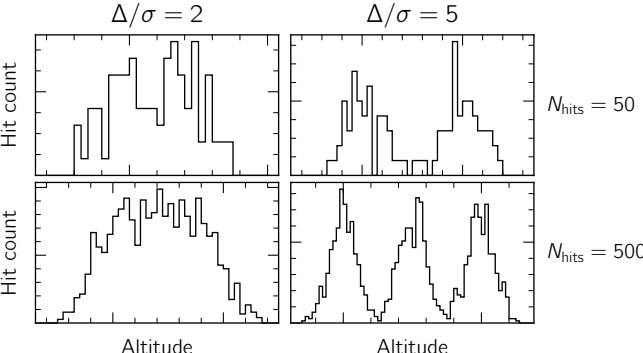

**Figure A1.** Simulations of 2 (top row) and 3 (bottom row) cloud layers using random normal distributions with $N_{\text{hits}}$ artificial cloud base hits per layer. Individual layers are separated by $\Delta = 2$ (left column) and $\Delta = 5$ (right column) standard deviations $\sigma$.

The Akaike Information Criterion ($AIC$) and Bayesian Information Criterion ($BIC$) scores provide means to assess the goodness of fit of different Gaussian mixture models. The smaller the $AIC$ or $BIC$ scores, the better the model fit. In ampycloud, we use $BIC$ scores to decide whether a given group $\mathcal{G}$ is composed of sub-layers. The $AIC$ score tends to favor models with larger numbers of components, which is at odd with the ampycloud approach of favoring –for near-equal scores– the least number of Gaussian components to prevent unjustified sub-layering. Therefore, we rely on the $BIC$ score only.

The relative likelihood of different Gaussian mixture models can be used to assign probabilities to each of them. The Bayesian probability of model $i$ is:

$$p_{BIC,i} = \frac{e^{-0.5(BIC_i - BIC_{\min})}}{\sum\limits_{i} e^{-0.5(BIC_i - BIC_{\min})}} \tag{A1}$$

with $BIC_{\min}$ the minimum $BIC$ score among the Gaussian mixture models under consideration.


We show in Fig. A2 the values of $p_{BIC,i}$ for the simulated datasets with 2 Gaussian sub-layers. The Gaussian mixture model approach appears incapable of identifying 2 distinct components when $\Delta \lesssim 2\sigma$, i.e. when the layers are separated by less than twice their standard deviation. For cases with $\Delta \gtrsim 2\sigma$, the probability of the distributions being comprised of a single Gaussian distribution drops extremely rapidly, in favor of the 2 components model. The transition is slightly less sharp for cases with

few number of cloud base hits (i.e. cases where $N_{\text{hits}} \lesssim 50$). The case of 3 simulated cloud layers is shown in Fig. A3. A single component is favored by the Gaussian Mixture Model approach for cases where $\Delta \lesssim 2\sigma$. A narrow, intermediate zone favoring 2 components is present for cases where $2\sigma \lesssim \Delta \lesssim 3\sigma$, whereas 3 components are correctly identified for cases where $\Delta \gtrsim 3\sigma$.

Unlike the simulated cases presented here, real cloud base hits are not following a Gaussian distribution, in particular due to

temporal trends in the cloud base heights (a fact which is clearly visible in Figs. B1 to B8). For real cases, using the probabilities $p_{BIC}$ to select the optimal number of sub-layers present inside a given group $\mathcal{G}$ works *too efficiently*, in the sense that a single Gaussian component is ruled out too rapidly in the case of broad, flat layers. To circumvent this limitation, ampycloud uses a slightly-adjusted selection criteria based on the $BIC$ scores, which varies more smoothly as a function of the (normalized) layer separation $\Delta/\sigma$. The general idea is to assume 1 component is present, and only favor a solution with 2 (or 3) components

if the decrease of the associated $BIC$ scores is sufficiently *significant*. In other words, a model with $j$ components is favored over a model with $i$ components only if:

$$BIC_j < \delta \cdot BIC_i \tag{A2}$$

where $\delta$ is a multiplicative factor (a parameter of the ampycloud algorithm).

The consequences of these selection criteria are shown in the bottom rows of Fig. A2 and A3. Varying the value of $\delta$ allows to more easily decide the level $\Delta/\sigma$ at which 2 (resp. 3) components are favored over a single one. With the default ampycloud value of $\delta = 0.95$, this transition occurs at $\Delta/\sigma \approx 5$. Sub-layers are therefore identified in a given group $\mathcal{G}$ only if they are well separated, as illustrated in Fig. A1. For the case of 3 simulated sub-layers, the selection criteria defined in Eq. A2 also has the advantage that it never favors 2 components (unlike the probabilities $p_{BIC}$, as illustrated in Fig. A2). If 3 components cannot

be identified unambiguously, no sub-layering occurs.

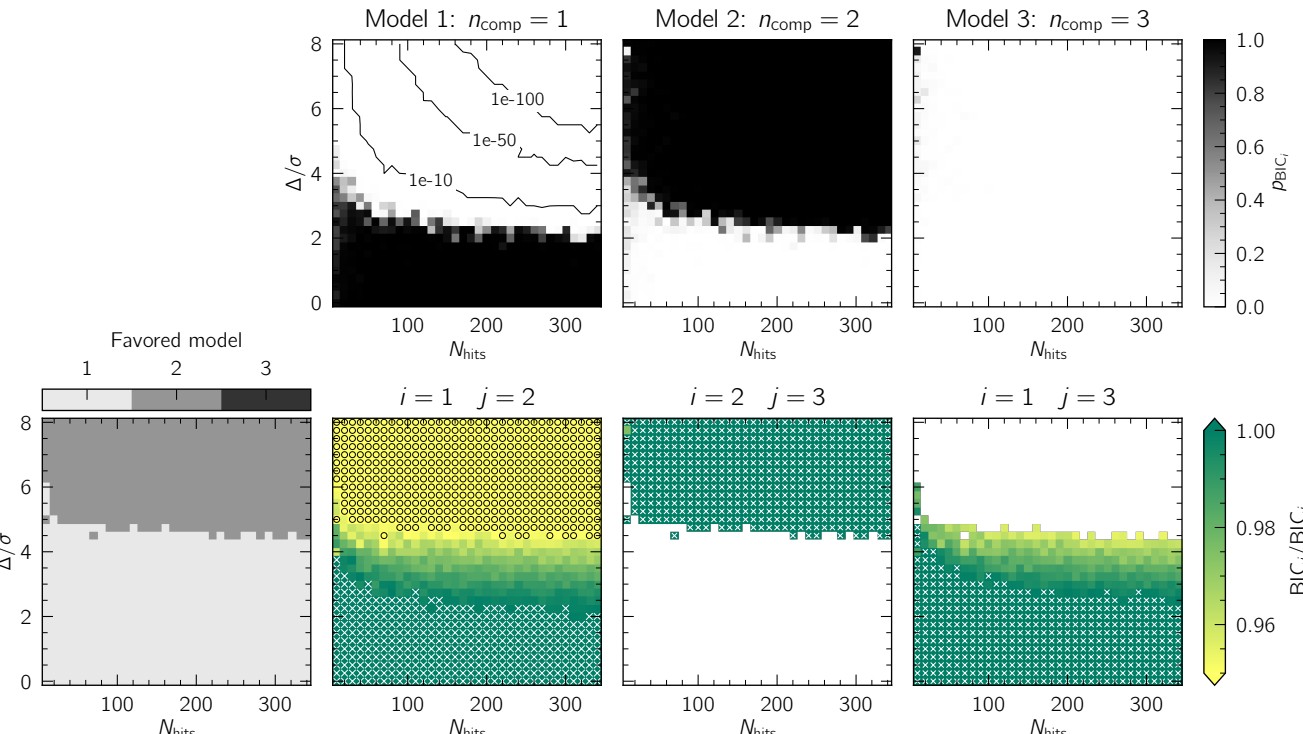

**Figure A2.** Illustration of the ability of the ampycloud layering step to distinguish the presence of 2 simulated Gaussian sub-layers inside a group $\mathcal{G}$, as a function of the relative separation of the sub-layers $\Delta$ (expressed in terms of the layers standard deviation $\sigma$) and the number of cloud hits per layer $N_{hits}$. Top row: $BIC$ probabilities $p_{BIC}$ computed following Eq. A1, built from the relative likelihood of Gaussian mixture models with 1 (left), 2 (middle), and 3 (right) components. Each pixel in the image corresponds to the median of 10 independent realizations. For very low numbers of hits per layers ($N_{hits} \lesssim 50$), the selection of the best Gaussian mixture model is less sharp. When the sub-layers are close from one another (with $\Delta \lesssim 2\sigma$), the Gaussian mixture model approach is unable to reliably identify two distinct components. Bottom (right): distributions of $BIC_j/BIC_i$, for the cases of 1 component vs 2, 2 components versus 3, and 1 component versus 3. Cases where $BIC_j/BIC_i > 1$ are tagged with a white cross: these are regions where the model with $j$ components is unambiguously rejected against the model with $i$ components because of a larger $BIC$ score. On the other hand, cases where $BIC_j/BIC_i < 0.95$ are tagged using black circles. For these and for these only, ampycloud will favor the model with $j$ components following Eq. A2. The resulting map of the model favored by ampycloud as a function of $N_{hits}$ and $\Delta/\sigma$ is visible in the bottom-left diagram.

## Appendix B: Detailed examples

We present in Figs. B1 to B12 the ampycloud diagnostic diagrams from a series of representative, real situations taken from the LSGG and LSZH aerodromes. These examples serve to illustrate the behavior of ampycloud in different conditions, and compare the algorithm's output with the official METAR cloud codes that were validated and issued by an AMO at the time. For the ease of readability, each example is discussed in detail in the associated figure caption. These examples are all part of


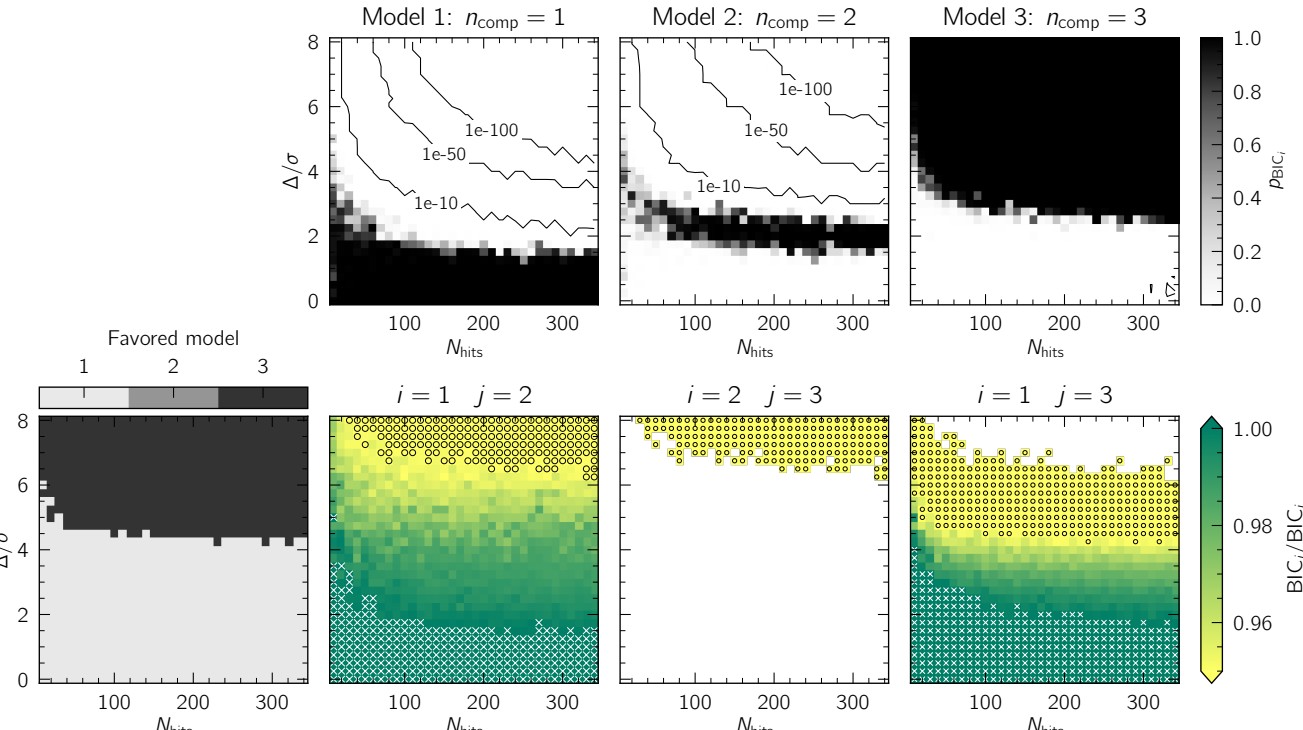

**Figure A3.** Same as Fig. A2, but for 3 simulated layers. As for the case of 2 simulated layers, the ampycloud layering step is able to correctly identify 3 distinct layers when they are separated by ~5 standard deviations ($\Delta \gtrsim 5\sigma$). A single layer is favored otherwise.

the set of cases used to verify the scientific behavior of the ampycloud algorithm over the course of its development, by means of dedicated tests (ampycloud, 2024e) run using the pytest module.

The underlying sets of cloud and VV hits associated to each example are made available to the interested reader, alongside
a small Python script designed to process them using ampycloud and generate the associated ampycloud diagnostic diagrams. This material is archived on Zenodo and publicly available (under a Creative Commons Attribution 4.0 International license; Vogt, 2024).

*Author contributions.* The ampycloud algorithm was designed by Vogt, with inputs from Foresti and Regenass. The ampycloud Python package was assembled by Vogt with important contributions from Regenass, Foresti, Réthoré and Tarin Buriel, and feedback from Bibby,
Juda, Balmelli, Hanselmann, du Preez and Furrer. The large scale statistical assessment of the code was performed by Regenass, with contributions from Foresti, Bibby and Tarin Burriel. All authors contributed to the writing of this article.

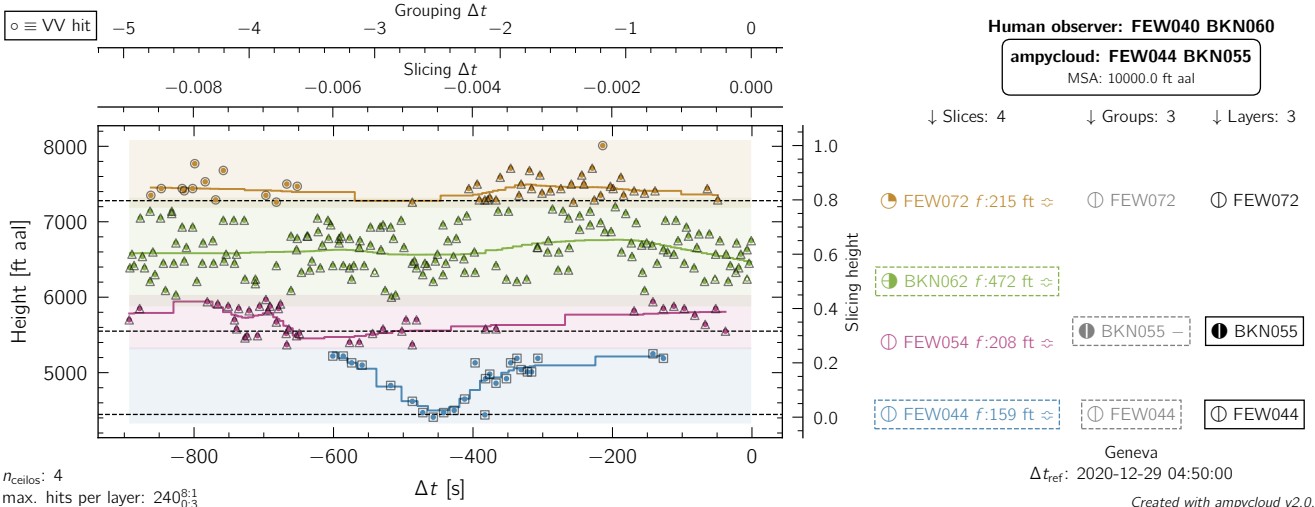

**Figure B1.** ampycloud diagnostic diagram for LSGG on 2020-12-29 at 04:50 UTC. The hits from 4 ceilometers over a period of 15 minutes are being considered. The four dominant slices are found to be overlapping, but the separations between individual hits is too large for the second processing step to bundle them all into a common master group. The symbol − in the Groups column on the right-hand-side indicates that sub-layers are being searched (but not found) by the layering step for the second group (from bottom; BKN055), which is the only structure with sufficient hits to do so (i.e. with a sky coverage fraction larger or equal to 2 oktas).

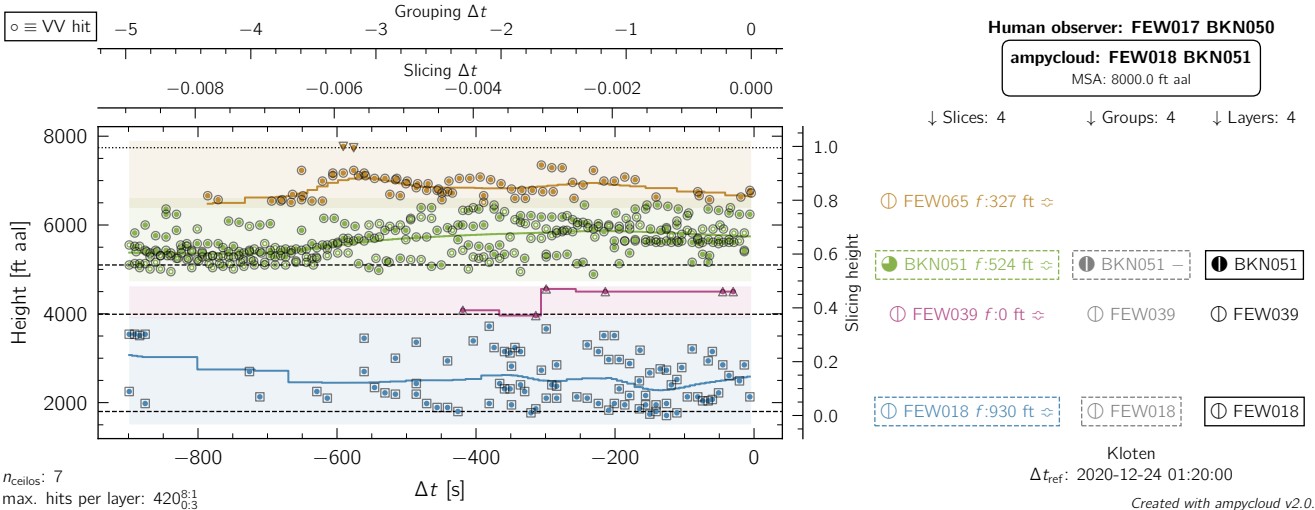

**Figure B2.** Same as Fig. B1, but for LSZH on 2020-12-24 at 01:20 UTC. VV hits are indicated using colored points with no filling: they are treated like regular cloud base hits by ampycloud. In this example, the grouping step has been used to merge the majority of hits in the top two slices: only the highest two cloud base hits remain as the upper-most layer, but they are discarded as a minimum of 4 hits are required for a layer be considered having a sky coverage fraction above 0 oktas (i.e. $\Theta_0 = 3$).

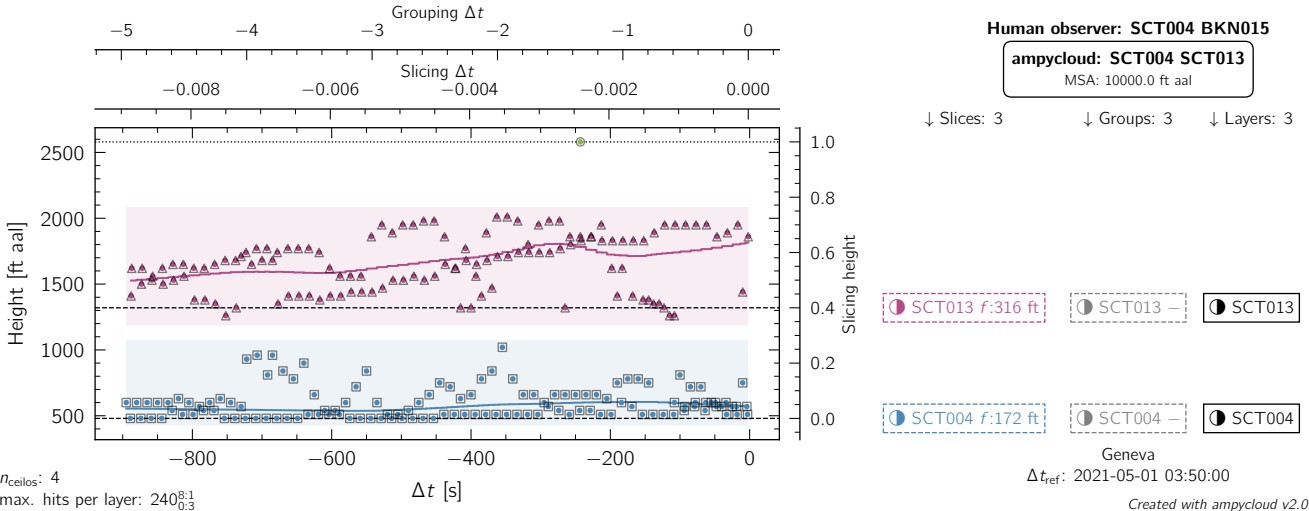

**Figure B3.** Same as Fig. B1, but for LSGG on 2021-05-01 at 03:50 UTC. In this case, the slicing step correctly identifies the two dominant cloud layers present. The grouping and layering step do not modify the initial slices. The top SCT013 layer slightly underestimates the sky coverage fraction that was reported to be BKN by the AMO, plausibly because of its obscuration by the lower layer.

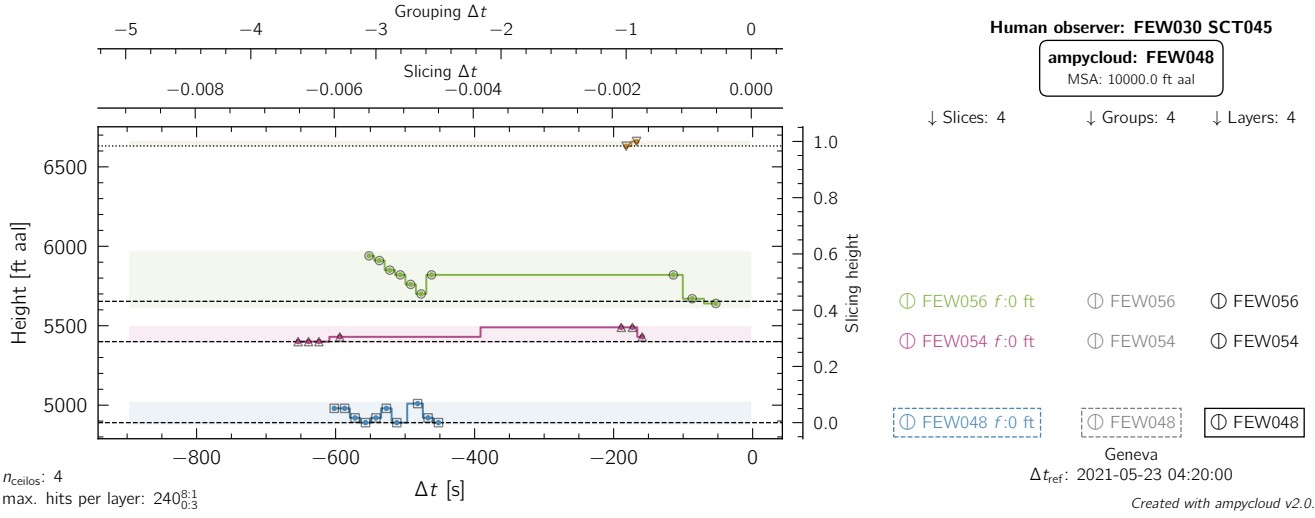

**Figure B4.** Same as Fig. B1, but for LSGG on 2021-05-23 at 04:20 UTC. In this case with very sparse layers, the slicing step correctly identifies the dominant cloud layers present. The layer at 3'000 ft reported by the observer is being missed entirely by the ceilometers over the duration of the time interval $\Delta t$=900 s, while the sky coverage fraction of the layer at 4'500-4'800 ft is being slightly underestimated by the ceilometers. Should a user prefer a less granular output at high altitudes, it is sufficient to change the parameters $\Delta h_{l,\mathrm{vals}}$ and $\Delta h_{l,\mathrm{lims}}$ to set a minimum separation value of 500 ft above 5000 ft, for example.

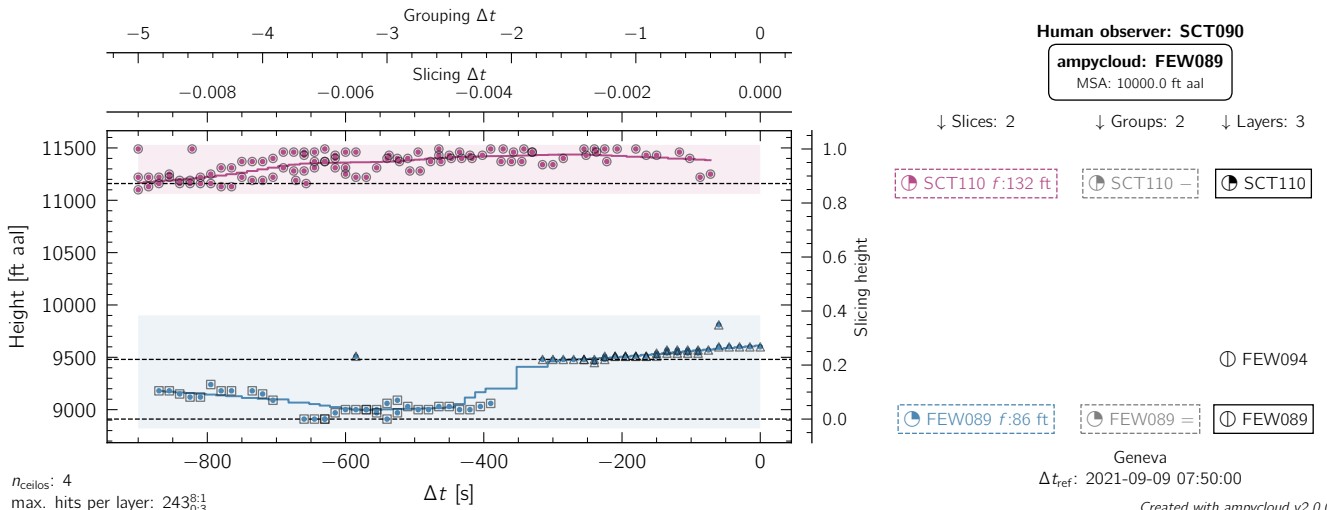

**Figure B5.** Same as Fig. B1, but for LSGG on 2021-09-09 at 07:50 UTC. The ampycloud layering step identifies two sub-layers separated by ∼500 ft in the bottom group, with a sky coverage fraction of FEW smaller than the SCT090 reported by the AMO, who likely merged the two sub-layers together. It is however worth noting that even the group FEW089 identified by ampycloud (bottom entry of the middle column) does not reach a sky coverage fraction of SCT, indicating that the ceilometers were globally looking at clear sightlines. The top layer SCT110 detected by ampycloud is not reported in view of the MSA applicable at LSGG.

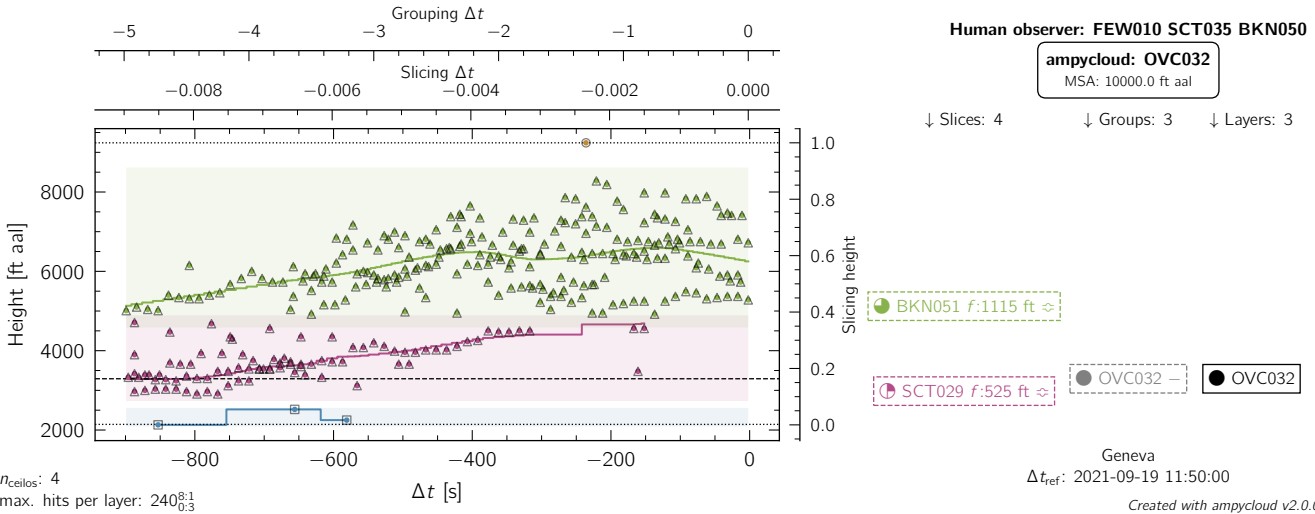

**Figure B6.** Same as Fig. B1, but for LSGG on 2021-09-19 at 11:50 UTC. The AMO reported two distinct layers at 3'500 ft and 5'000 ft, consistent with the initial ampycloud slices. The overall cloud hit distribution is however suggestive of a single coherent structure increasing from 3000 ft to 5'000 ft over the 15 min interval, and is identified as such by the grouping step of the ampycloud algorithm. The bottom layer at 1'000 ft is missed entirely by the ceilometers over the duration of the time interval $\Delta t = 900$ s.

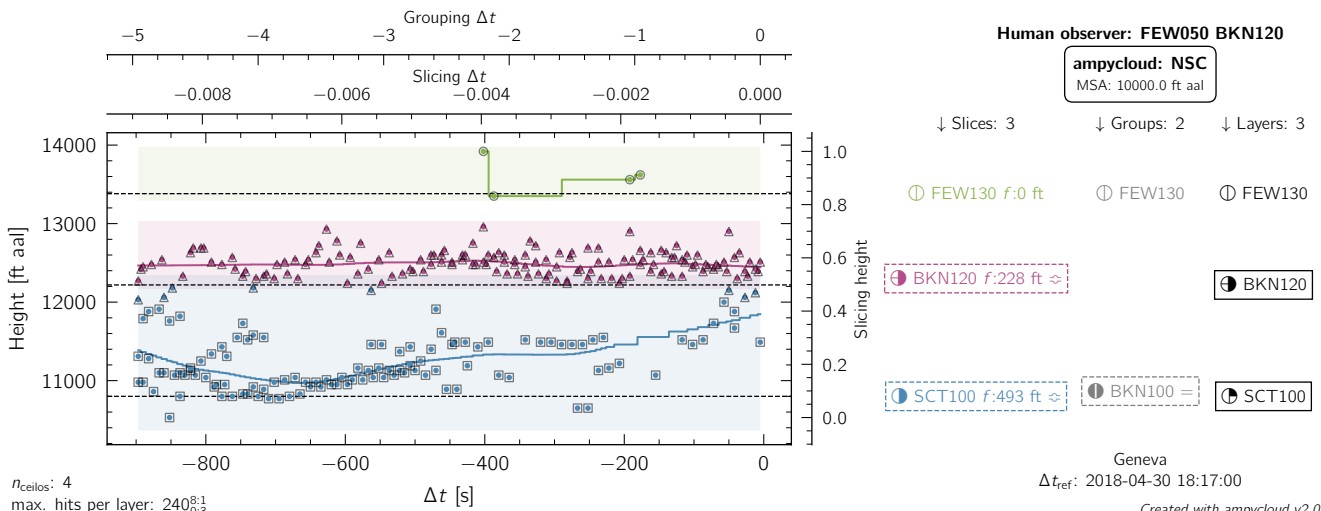

**Figure B7.** Same as Fig. B1, but for LSGG on 2018-04-30 at 18:17 UTC. The value of $\Delta_{MSA}$ is exceptionally set to 4000 ft in this example, for visualization purposes. The bottom two slices are merged into a single structure by the grouping step (as they connect to each other at the start and end of the interval), and are then re-separated into two distinct layers. Clouds at 5'000 ft were missed entirely by the ceilometers over the duration of the time interval $\Delta t$=900 s. This example also illustrates the difficulty in blindly comparing METARs with AUTO METARs. The AMO decided to report the BKN120 layer despite the MSA applicable at LSGG (whereas ampycloud simply ignores it) leading to an (apparently) missed ceiling.

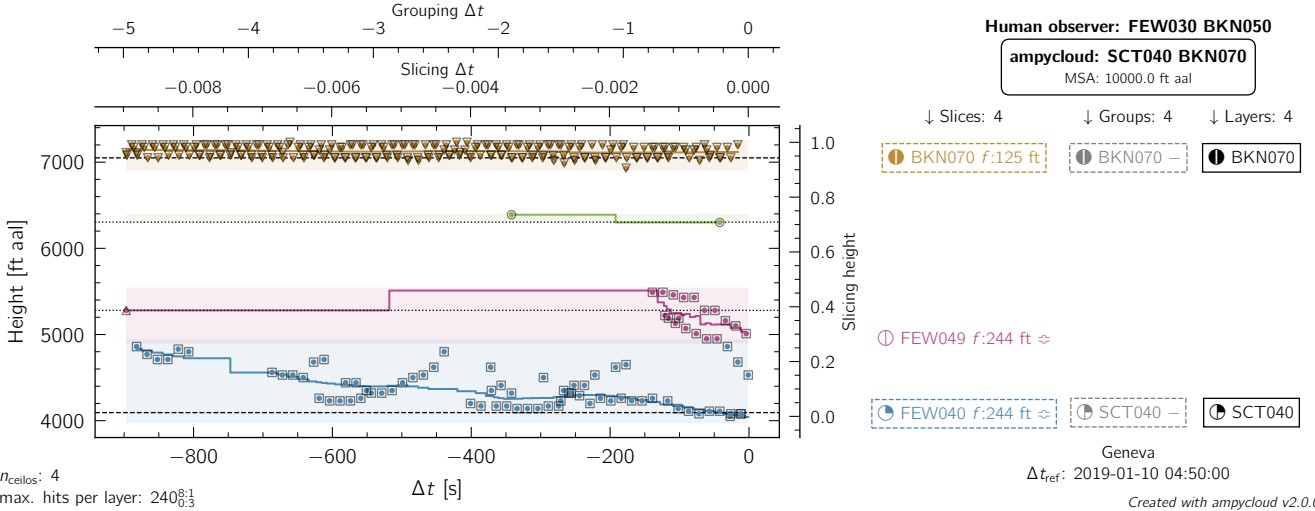

**Figure B8.** Same as Fig. B1, but for LSGG on 2019-01-10 at 04:50 UTC. It is because ampycloud accounts for the fluffiness of the bottom two slices that the grouping step decides that these form a single structure, despite the somewhat offset cluster of hits appearing within the last 150 s. It is not clear why the layer at 7'000 ft was not reported in the METAR, while clouds at 3'000 ft were missed entirely by the ceilometers over the duration of the time interval $\Delta t$=900 s. **29**

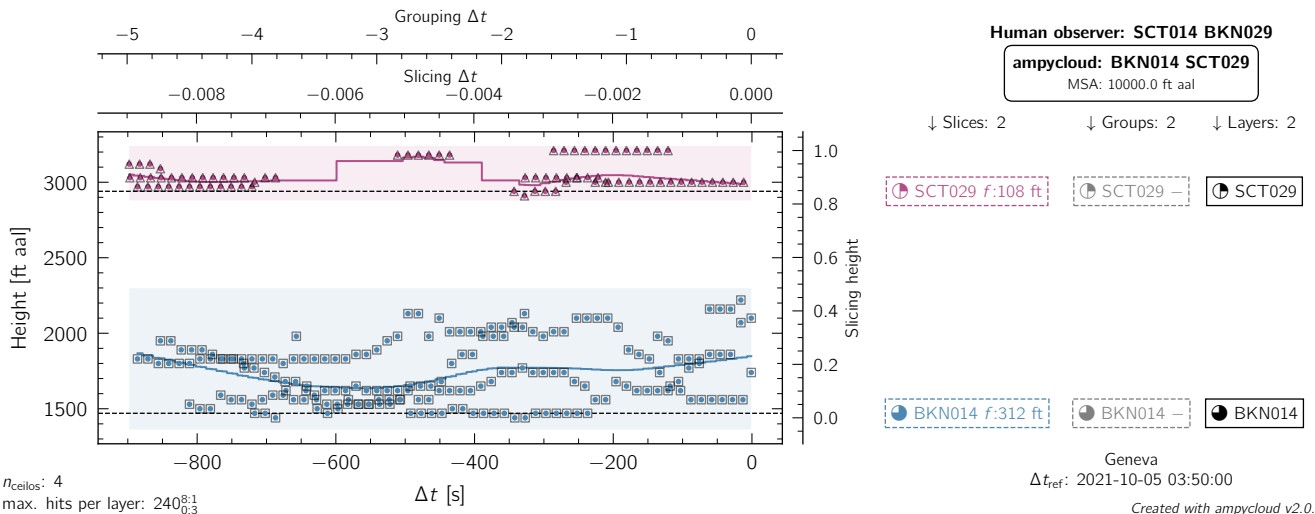

**Figure B9.** Same as Fig. B1, but for LSGG on 2021-10-05 at 03:50 UTC. The slicing step correctly identifies the two cloud layers present, albeit with a slightly different sky coverage fraction than those reported by the AMO. The top group is not being separated in 2 distinct sub-layers by the layering step because their separation of $\sim$180 ft would be smaller than the minimum value of $\Delta h_{l,\mathrm{vals}} = 250$ ft.

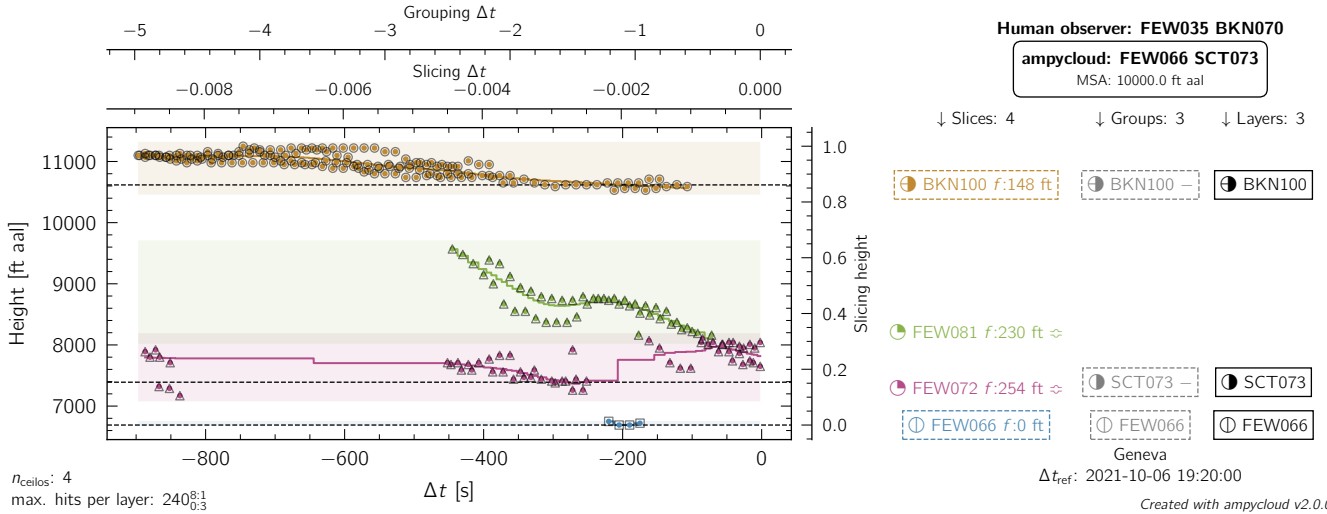

**Figure B10.** Same as Fig. B1, but for LSGG on 2021-10-06 at 19:20 UTC. The grouping step is key to connect the 2 central slices given the rapidly decreasing nature of the cloud base between $-400$ s and $-100$ s. Once again, the layer FEW035 reported by the AMO is being missed entirely by the ceilometers over the duration of the time interval $\Delta t$=900 s.

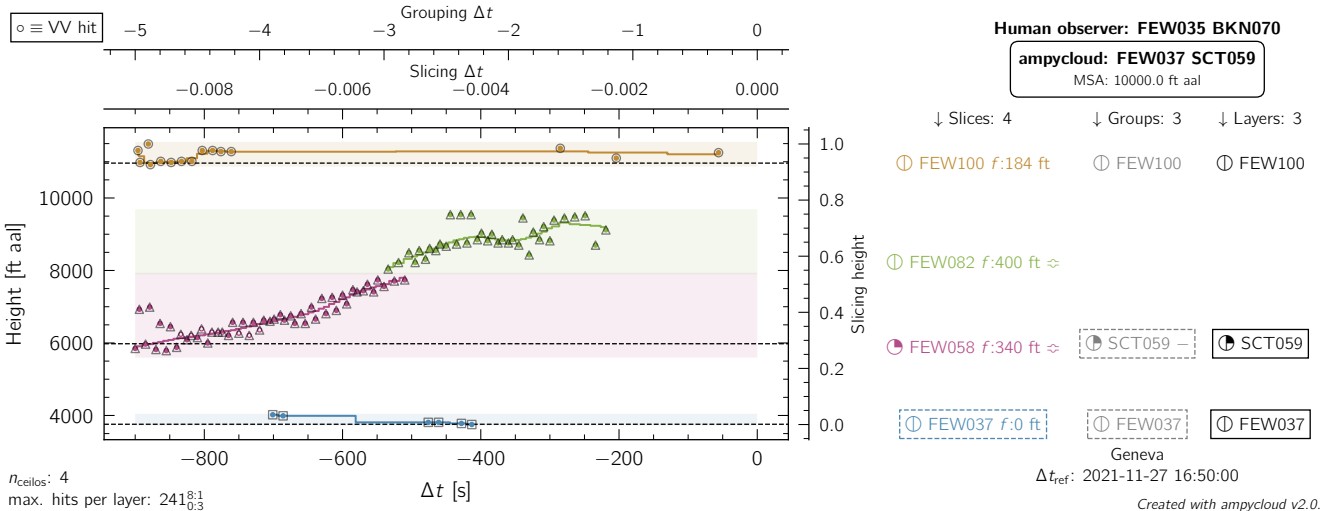

**Figure B11.** Same as Fig. B1, but for LSGG on 2021-11-27 at 16:50 UTC. The grouping step correctly joins the middle two slices in this case with a rapidly rising cloud base. ampycloud considers the full set of hits (with $\beta_t = 100\%$ by default) to derive the base height of this layer, whereas the AMO likely ignored the oldest ceilometer data.

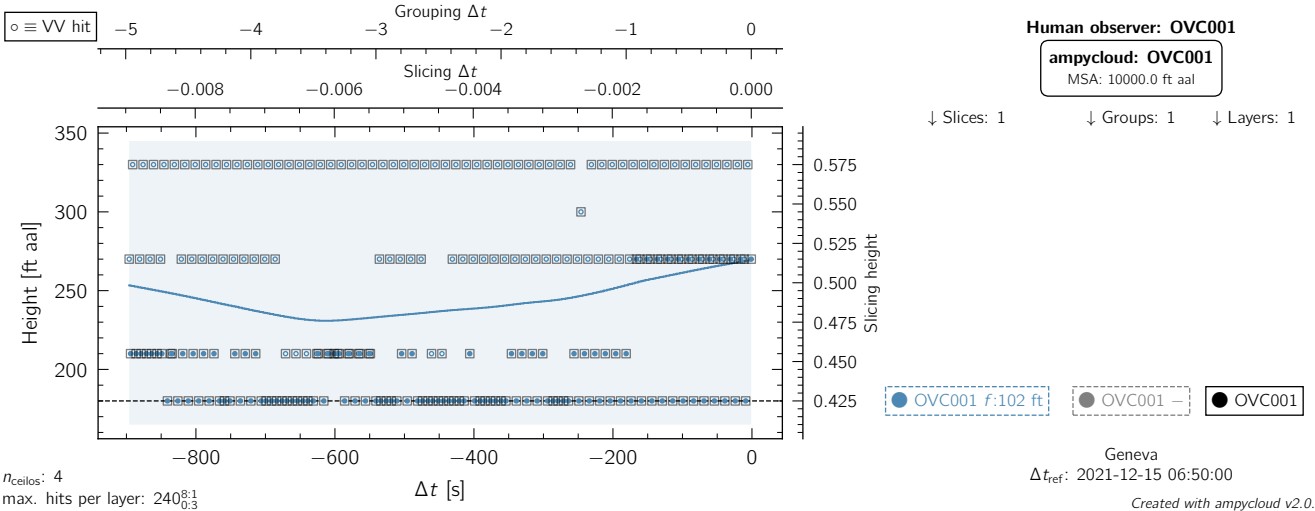

**Figure B12.** Same as Fig. B1, but for LSGG on 2021-12-15 at 06:50 UTC. A single OVC001 slice is (correctly) identified by the ampycloud algorithm in this case with a large number of VV hits reported by the ceilometers. We stress however once again that ampycloud is not intended nor designed to formally decide whether a VV code must be issued in the AUTO METAR (instead of OVC001 in this example). This task should be performed by a separate algorithm focusing on vertical visibility detection and reporting.

*Competing interests.* The authors declare no competing interests.

*Acknowledgements.* We thank Kenneth Boutin and Chet Schmitt for sharing with us the detailed description of the latest sky-condition algorithm used by NOAA's ASOS Program. We are grateful to the two anonymous reviewers and P. Kuma for their feedback, which allowed us to improve this article in several parts. Diagrams in this article have been generated using the ampycloud Python module, which relies on the following Python packages: matplotlib (Hunter, 2007), numpy (Harris et al., 2020), pandas (McKinney, 2010; The pandas development team, 2021), scikit-learn (Pedregosa et al., 2011), scipy (Virtanen et al., 2020), and statsmodels (Seabold and Perktold, 2010). The ampycloud diagrams were enhanced using the metsymb LaTeX package (Vogt, 2023).

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
