# Peer review of "ampycloud: an open-source algorithm to determine cloud base heights and sky coverage fractions from ceilometer data"

_Atmospheric Measurement Techniques, 2023_

## Referee Comment (RC1)

**Reviewer's comments**

**General comments**

The preprint entitled *"ampycloud: an algorithm to characterize cloud layers above aerodromes using ceilometer measurements"* has been reviewed for possible publication in Atmos. Meas. Tech. The authors developed a new algorithm and associated Python package claiming it is suitable for the fully automated reporting of cloud information relevant for Meteorological Aerodrome Report (METAR) at Swiss civil aerodromes. The open source strategy with the Python package and its online documentation are positive, but the potential user group is likely limited to other weather services.

The manuscript is clearly organised, although individual paragraphs and figures belong in other sections or in the appendix. The diagnostic diagrams are really nice, although they do require a certain amount of background knowledge. The writing style is well understandable, even if some issues could have been explained better with more appropriate wording. This would also avoid unnecessary repetition of the same content both in the main sections and in the figure captions. Spelling and grammar are appropriate but need improvement and British English should be used consequently. Occasionally, the authors become entangled in contradictions and misunderstandings are inevitable for less knowledgeable readers. In a few places there is no corresponding reference, e.g. to the CL31 manual from *Vaisala* (2009).

Essentially, it is a matter of assigning the cloud hits originating from $n$ ceilometers, which are derived from the actually measured backscatter signal, to cloud clusters, summarising these as far as possible into cloud layers and finally weighing up which of these are relevant for reporting, considering all the requirements and rules of the *ICAO* (2018). The authors use interesting approaches applying agglomerative clustering method and Gaussian mixture model, but it seems like they take a sledgehammer to crack a nut. In contrast, some key requirements of the *ICAO* (2018) are not met like reporting "NSC" (nil significant cloud) or vertical visibility (VV) with altitude value (see specific comments).

The quality of any similar cloud algorithm depends strongly on the performance of the ceilometer type used. Various ceilometers from the same or different manufacturers derive from their attenuated backscatter profile different cloud and VV hits at a given time and each device has its own strengths and weaknesses (e.g. *Illingworth et al.*, 2019). `ampycloud` is designed and tuned for CL31 and needs to be adapted or even extended if you replace the ceilometer. In the specific comments, I try to explain the limitations of the CL31 with regard to its use in aviation based on personal experience.

From the results presented, I cannot recognise any superiority of `ampycloud` over existing similar cloud algorithms, except that the implementation here is significantly more complex. Furthermore, its assessment based on observations is not convincing. Contrary to what the authors call an improvement of the algorithm, it would be an actual innovation with potential to be published, if a) the integration time has been made dependent on the horizontal wind speed or b) the measured backscatter signal has been used to derive cloud hits and VV hits by yourself to get rid of the manufacturer black box algorithms. In my opinion, this manuscript is not suitable for publication in its current form.

**Specific comments**

My recommendation for the title is:

> "`ampycloud`: *an algorithm for determining the amount and base altitude of cloud layers over aerodromes using ceilometer data*"

On the one hand, there are numerous macro- and microphysical cloud properties (e.g. *Gao et al.*, 2014; *Luebke et al.*, 2022), but `ampycloud` is only able to determine cloud amount and cloud base altitude based on the data of *n* ceilometers. On the other hand, the quantity actually measured by ceilometers is the backscatter signal (attenuated backscatter), while the cloud or VV hits are derived by application of manufacturer-specific algorithms due to the lack of a quantitative definition of a cloud (section 2.1.1 of *WMO*, 2017).

`ampycloud` is a strongly scientific rather than application-orientated implementation. There is no need to determine cloud amount and cloud base altitude with best accuracy, considering that for the METAR/SPECI the former is classified into FEW, SCT, BKN, OVC and the latter is rounded down to 100 ft increments below 10,000 ft altitude. The current version of `ampycloud` is only suitable to a limited extent for the complete, operational determination and reporting of cloud and VV information in METAR/SPECI or MET REPORT/SPECIAL syntax, i.e. major revisions are required:

- NSC is currently not considered but required by section 4.5.4 of *ICAO* (2018). Your Fig. B7 is just such a case where you have to report "NSC" in the METAR because there are clouds (above MSA) but they have no operational significance.

- Height/altitude values for VV can currently neither be determined nor reported although it is a requirement according to sections 4.5.4 and 4.6.5.1 of *ICAO* (2018). The CL31 provides VV hits and I wonder why the authors argue that another algorithm should handle it. Your data in Fig. B12 could also mean a so-called VV case, which leads to the possible result "VV001" in the METAR.

- Your line of argument for using a shorter integration time of $\Delta t$ = 15 min is flawed. Such a short time period does not necessarily require a weighting of the latest cloud hits in order to better react to rapid changes in the cloud field, but e.g. for 30 min it makes perfect sense. METARs are representative for 8 km (even 12 km for convective clouds) radius around the airport reference point and the reporting time is 30 min. *Nadolski* (1998) pointed out that "*… a 30-minute time period provided an optimally representative and responsive observation similar to that depicted by an observer*". Do you have observational evidence that your choice is better? Depending on the wind direction, the ceilometer detects clouds in the downstream half-space, i.e. only those clouds that have passed over the device. The crucial problem with a shorter $\Delta t$ is the loss of information, which leads to less agreement with the observer who is looking at the entire half-space.

- I wonder why the authors explain `ampycloud` using Fig. 1, where obviously an integration time of $\Delta t$ = 20 min (1200 s) was used, but they only speak of $\Delta t$ = 15 min (900 s) both in the main text and in the figure captions of the appendix. This is an inconsistency and leads to misunderstandings. Why did you switch from 20 minutes to 15?

- Your slicing step applies the Manhattan metric as a distance function and average linkage as a fusion method. Rescaling the time axis is a clever hack to reduce the 2-D problem to 1-D, but why are you still using a 2-D clustering method that is computationally less efficient and slower? The original (historical) implementation of the `ampycloud` algorithm is no reason to deny further developments. From your initial slicing step, which is similar to the binning and clustering steps of the ASOS (*Nadolski*, 1998) you have generated two new problems: 1. To combine two cloud clusters that were incorrectly assigned to different slices but actually belong to each other you use an alternative distance function (Euclidian metric) and fusion method (single linkage). In the latter outliers often give rise to chain effects ("single-link effect") and tiny clusters that consist of only a few elements. 2. The problem that two independent

cloud clusters were incorrectly assigned to the same slice you try to solve using Gaussian mixture models. Wouldn't it be more plausible to use Gaussian mixture models to combine cloud clusters to form final cloud layers? Models with lower BIC are generally preferred but a lower BIC does not necessarily indicate one model is better than another. Have you applied other distance functions and fusion methods for the slicing and grouping steps to be sure what is the ideal choice? Is the Gaussian distribution really suitable for cloud hits in the layering step? I doubt these questions have been adequately answered.

- The lack of comparison with the results of at least one similar cloud algorithm already in operation. The authors claim in section 4 that this *"… is outside the scope of this article."*, but they need to show the added value of `ampycloud` to justify the higher computational effort and complexity.

- In section 3.4 the authors mention to use *"a reference dataset of 2128 cases extracted over a 5 year period (2018-2022) from LSGG METARs"* for the statistical evaluation and assessment of `ampycloud`. Even if SPECIs are not considered, a single year comprises 17520 METARs (twice an hour for 365 days). On the other hand, 2128 cases correspond to about 44 days. In relation to the 5-year period, it is only slightly more than 2.4 %. From a statistical point of view, this is definitely insufficient, your results are therefore not significant and the pre-selection of cases harbours the risk of concealing serious shortcomings in your algorithm.

- There is neither a controversial discussion nor at least a citation to the strengths and weaknesses of the observer, who is not the "ground truth" in all situations. This must be considered when discussing the results of `ampycloud` with regard to a possible under- or overestimation.

- The authors only present results for ideal input data, but this does not always apply to fully automatic operation. Please provide more detailed insights into the exception handling. How does your algorithm react if …
  - there is a "complete failure", e.g. due to power or network failure, and no input data is (temporarily) available?
  - there is a failure of individual or multiple ceilometers (e.g. due to maintenance)?
  - a single ceilometer measures but transmits an error code?

- Some important information about the CL31 is missing, e.g. the output interval of 15 s (4 hits per minute), which can only be inferred from the *max. hits per layer* in the diagnostic diagrams with some difficulty.

The Vaisala CL31 algorithm derives from the actual measured backscatter signal either up to three cloud hits in different heights or a value for vertical optical range (VOR) above ground level at any given time. Similar to determining cloud base altitude VV hits should be used to derive an altitude value for VV. By definition, VOR is the height above the ground up to which the extinction coefficient must be integrated in order to reach a certain threshold value, usually around 3 (*Werner et al.*, 2005). VOR is always higher than all cloud hits detected at a given time, because a well-performing ceilometer can always "see" a little into optically thick clouds and even through optically thin clouds.

However, Vaisala obviously uses a significantly smaller threshold value in order to have an alternative definition for the derivation of cloud hits. They call it a VV hit, but it's basically a cloud hit. Figures B1 and B2 show very nicely that a) cloud hits and VV hits merge almost seamlessly into one another and b) `ampycloud` would be completely wrong if the VV hits were not considered cloud hits. The crucial problem is that these two quantities are completely different things from a physical/meteorological point of view.

In the following there are specific comments to certain lines or figures of the preprint:

ll 24     MET REPORT/SPECIAL is not representative of the runway, but of the threshold. Each runway has 2 thresholds. Unfortunately, the MET REPORT/SPECIAL syntax is e.g. "... CLD RWY 05 OVC 800FT ...", which can lead to confusion.

| | |
|---|---|
| ll 84 | There are also parameters that have only computational or tuning relevance, such as your OVC/FEW thresholds or $\alpha_s$ in Tab. 2, which are arbitrarily chosen. |
| Fig. 1 | Figure 1 should be presented in section 2.4 if the abbreviations and parameters have already been introduced. The figure caption of Fig. 1 is very long. I prefer to describe the method and general information in the main text with reference to this figure. All figures in Appendix B and Fig. 1 should then be self-explanatory. |
| ll 120 | As you mention in relation to "nnn" in the caption of Fig. 1, a ceilometer does not provide vertical values relative to mean sea level (altitude = elevation + height), but it gives the height above ground level. The elevation corresponding to a specific location must be explicitly added to the settings. |
| lls 127 – 128 and 134 – 135 | *"Data are pooled together before the analysis, independently of the spatial distribution of the ceilometers."* and *"The specification of $h_{cid}$ for every hit is necessary to compute a correct estimation of …"* are contradictory statements. |
| ll 140 | What is meant with "signal range"? Is it the maximum range of detection? |
| Fig. 2 | Figure 2 should not be shown in the main sections, but at most in the appendix, as it is essentially just a repetition of the results from section 5 of *Wauben* (2002). The added value of $n > 1$ ceilometers for distinguishing BKN and OVC is rather limited, as both mean a ceiling in the end. |
| ll 160 | Meanwhile Vaisala provides an "airplane filter" and maybe it can even be utilized by CL31 with a firmware upgrade. |
| lls 262 - 263 | The triangles and rectangles in the blue slice in Fig. 1 reveal that $\alpha_s \cdot (h_{max} - h_{min})$ $\approx 0.2 \cdot (2{,}100\,\text{ft} - 950\,\text{ft}) = 230\,\text{ft}$. The distance between the two sub-layers is definitely larger than 230 ft. So why are they combined during the slicing step? |
| lls 350 – 351 | Or the differences are due to the observer's limited ability to distinguish between slightly more or less than 50 % cloud cover. |
| ll 357 | Lower cloud bases occur almost twelve times (24.6 %) as often as higher ones (2.1 %). Therefore, it is not a tendency but systematic. This is potentially problematic when applying the SPECI criterion in section 2.3.2 f) of *ICAO* (2018). |
| ll 360 | Or your $\Delta t$ is too short and you miss some clouds that passed over CL31 earlier. |
| Fig. B4 | Perhaps it is the same cloud layer that has passed over at least 2 ceilometers with a time delay. Then the two cloud clusters (green and red) should be combined into a single cloud layer. This would be more consistent with the 2nd cloud group reported by the observer. |
| Fig. B5 | The cloud hits around 9,100 ft and 9,500 ft belong definitely to a single cloud layer, as it is a similar situation as for the red slice in Fig. B6. All hits together would likely result in SCT. |
| Fig. B6 | The cloud hit distribution and your slicing step clearly speak in favour of 3 cloud layers similar to the observer-based METAR meaning that the grouping and layering steps fail. |
| Fig. B10 | Perhaps your $\Delta t$ is too short and you therefore cannot reproduce the clouds at around 3,500 feet. |

**Technical corrections**

| | |
|---|---|
| lls 1, 15, 45, … | Either "sky coverage" or "cloud amount" but use a consistent wording. |
| ll 28 | I would prefer "international" rather than "civil" airports, as regional airports as a different category are not limited to goods and trade. |
| ll 29 | "in addition" instead of "an additional" |

| | |
|---|---|
| ll 59 | "cloud layers" without 2nd "s" |
| ll 61 | "its performance" singular is enough |
| lls 66 – 71 | This paragraph should be moved to section 2 or even section 3. |
| lls 98 – 99 | Move these two sentences and Fig. 1 to section 2.5 and shortly refer to it here avoiding repetition of the same content. |
| ll 111 | Use "cloud and VV hits" instead of "ceilometer hits". |
| Fig. 1 | - $\Delta t = 20$ min should be the same as for figures in Appendix B (15 min)
- omit sup-/subscripts and explanation of OVC/FEW thresholds as these information can also be found in Tab. 2.
- which airport is shown? (MSA: 8,000 ft refers to LSZH but $n_{ceilos}$: 4 to LSGG)
- max. hits per layer: 160 is incorrect
$n_{ceilos} \cdot$ hits/min $\cdot \Delta t = 4 \cdot 4 \cdot 20 = 320$ (doesn't fit) but $= 2 \cdot 4 \cdot 20 = 160$ (fits) |
| ll 117 | Use "cloud hit" instead of "ceilometer hit". |
| ll 120 | "derived cloud base height above ground level" or "derived cloud base altitude above mean sea level" but no mixture and not including "measured" |
| ll 124 | "backscatter profiles" |
| Tab. 1 | Maybe add the CL31 output interval of 15 s, the elevation, MSA, etc. |
| ll 148 | - use "time interval", "time increment" or "integration time" but not "length"
- "can be bundled" instead of "can be pooled" |
| ll 151 | "… time difference relative to the latest." |
| lls 157 – 159 | "Assuming *max. hits per layer* is 160, then we use a threshold value of 99.375 % for OVC and 1.875 % for FEW, accepting 1 hole for the former and requiring 4 hits for the latter." Your wording is far too cumbersome. Why do you apply an implicit dependency on *max. hits per layer* instead of defining fixed percentages? |
| lls 178 – 179 | "This distance threshold is $\Delta h_l = 250$ ft between 0 – 10,000 ft (1,000 ft above 10,000 ft)." Your wording is rather cryptic. |
| lls 186 – 187 | such repetitions should be avoided |
| Tab. 2 | - "behaviour" in the figure caption
- table too long and covers page number
- add the meaning of parameters in a separate column and omit Python variable |
| ll 206 | misleading footnote (could be power 3) |
| lls 222 – 223 | "By (our) definition, slices *i* and *j overlap* if *one* of the following two conditions *is true/met*:" |
| ll 227 | use "For example, the two top slices in Fig. 1 (green and red) overlap, …" |
| ll 241 | use "the latter value" instead of "this value" |
| ll 250 | A reference to Tab. 2 is missing for $L_{frac}$ and $L_{it}$ |
| ll 271 | "overestimation" instead of "overestimate" |
| ll 276 | use precise wording to avoid "In other words, ..." |
| ll 277 | use "assumes" and "number" instead of "will deem" and "amount", respectively |
| ll 300 | Avoid text extending beyond the edge of the page. |
| ll 304 | the correct reference is "(ampycloud, 2024a) |
| ll 334 | use "5-year period" |
| ll 348 | use "The comparison of Fig. 3 and Fig. 4 shows, that false alarms …" |
| ll 355 | "height/altitude of the cloud layer with the highest degree of coverage" instead of "densest cloud layer" |
| ll 357 | "than" instead of "that" |

| ll 359 | consistently use either "cloud base height" or "cloud base altitude" |
| Fig. 5 | maybe use "hft" (hecto feet) instead of "100 ft" |
| ll 416 | "the accompanying material" instead of "associated supplementary material" |
| lls 426 – 428 | This information does not belong in the conclusions. |
| ll 487 | "figure caption" |
| Figs. B5, B11 | *max. hits per layer* must be 240 |

**References**

Gao, W., C.-H. Sui, and Z. Hu (2014): A study of macrophysical and microphysical properties of warm clouds over the Northern Hemisphere using CloudSat/CALIPSO data, *J. Geophys. Res. Atmos.*,119, 3268–3280, doi:10.1002/2013JD020948.

Illingworth, A. J., Cimini, D., Haefele, A., Haeffelin, M., Hervo, M., Kotthaus, S., Löhnert, U., Martinet, P., Mattis, I., O'Connor, E. J., and Potthast, R. (2019): How Can Existing Ground-Based Profiling Instruments Improve European Weather Forecasts? *Bulletin of the American Meteorological Society*, 100(4), 605-619. https://doi.org/10.1175/BAMS-D-17-0231.1.

International Civil Aviation Organization (2018): Meteorological Service for International Air Navigation, Annex 3 to the Convention on International Civil Aviation, 20th Edition, AN 3, ISBN 978-92-9258-482-5.

Luebke, A. E., Ehrlich, A., Schäfer, M., Wolf, K., and Wendisch, M. (2022): An assessment of macrophysical and microphysical cloud properties driving radiative forcing of shallow trade-wind clouds, *Atmos. Chem. Phys.*, 22, 2727–2744, https://doi.org/10.5194/acp-22-2727-2022.

Nadolski, V. L. (1998): *Automated Surface Observing System (ASOS) User's Guide*, Tech. rep., 112 pp. National Oceanic and Atmospheric Administration, Department of the Air Force, Federal Aviation Administration, and United States Navy, USA.

Vaisala (2009): *User's Guide Vaisala Ceilometer CL31*, Tech. rep., M210482EN-D, 142 pp., Vaisala, Finland.

Wauben, W. M. F. (2002): Automation of Visual Observations at KNMI: (II) Comparison of Automated Cloud Reports with Routine Visual Observations, in: Symposium on Observations, Data Assimilation and Probabilistic Prediction, AMS Annual Meeting, Orlando (FL), USA.

Werner, C., Streicher, J., Leike, I., Münkel, C. (2005). Visibility and Cloud Lidar. In: Weitkamp, C. (eds) Lidar. Springer Series in Optical Sciences, vol 102. Springer, New York, NY. https://doi.org/10.1007/0-387-25101-4_6.

World Meteorological Organization (2017): *International Cloud Atlas*, 2017 Edition, https://cloudatlas.wmo.int.

---

## Referee Comment (RC2)

This article presents a new algorithm: ampycloud, deployed by MeteoSwiss at different aerodromes within the country. As the authors state, given the opaqueness of ceilometer manufacturers' cloud layer height and amount algorithms, using these at a given aviation meteorological institute remains difficult. The article makes a cloud layer algorithm available, along with a suite of test cases for interested users to apply. The author present test cases and statistical studies in order to validate their algorithm. These, together with the algorithm deserve publication.

However, changes in the manuscript must be made in order to improve its clarity. Specifically, more explanation is needed for certain parts. For example, the algorithm is tunable through many parameters, yet, the authors do not explain the reason for their choices of various constants. Also, their choice of integration time is not well argued and it is not immediately clear why vertical visibility is not reported with ampycloud. These comments are detailed in the following sections

**General comments:**

Line 32: Please define auto metreport before using this name

Line 55-56: It also prevents combining different types of ceilometers. Combining different types of ceilometers might be done for e.g. financial reasons.

Introduction or section2:

Some background reasoning or framing on why ampycloud is designed the way it is-> It uses rather advanced statistical methods compared to for example the Larsson algorithm used by the KNMI (see e.g. Wauben et al. 2002). What are the advantages/disadvantages of ampycloud compared to e.g. the algorithm described by Wauben?

line 94: I find (pitfall 1) difficult to understand

Line 103: "Obscuration of the sky by fog or snow must be handled by a separate algorithm". Please clarify as to why this is the case.

Line 111: "relies on ceilometer hits". Perhaps a brief description of what a ceilometer hit is, together with a few words on how a ceilometer calculates cloud base heights would help the reader.

Line 120: what is meant by relative time (It is described later, but perhaps better to describe it here)?

Line 123-125: "One should note that for the moment, ampycloud cannot use full back-scatter profiles to derive cloud base hits independently from the ceilometers' proprietary software." Since deriving your 'hits' from backscatter profiles would just mean that you're clustering different cloud base hits, does this affect the quality of the algorithm?

Line 130: definition of AUTO METREPORT should be done earlier e.g. in the introduction

Line 132: Discussion Figure 2 should be in the main text.

Line 134-135: "The specification of hcid for every hit is necessary to compute a correct estimation of the sky covering fraction under the assumption that cloud layers have a unique base per time step per ceilometer line-of-sight.". Could you explain this further? Why is the hcid necessary for the correct estimation of the sky covering fraction given that all hits are treated equally, pooled together and sorted?

Line 149: a delta t of 15 and 6 minutes seems arbitrarily chosen: Could you provide more details as to why 15 and 6 minutes were chosen?

Line 155: I was under the impression that compliant METAR cloud layer codes were derived sequentially, meaning that first S, then G then L were computed in order to finally give up to 3 layers. With the current wording, it seems that a METAR code is generated for the 'S step', the 'G step' and the 'L step'. Could you please clarify?

Line 156: Does that mean that a cloud layer is considered at the 4 hit? Does that mean that the cloud information other than NCD would be available to the AUTO METAR generation after 1 minute of detecting the beginning of a cloud layer (given a time resolution of 15 seconds for the ceilometer)? Could you say a few words on the operational suitability of this?

Line 170-172: "If a user were to prefer the cloud base heights to be more representative of the most recent hits within a slice/group/layer, the B_t parameter can be set to lower values, e.g. B_t = 30%. Note that changing the value of _t does not have any effect on the cloud amounts." As far as I understand, this is not the same as doubly weighing the most recent cloud hits. When doubling the most recent cloud hits, the oldest cloud hits still have a single weight, and so are still considered in an algorithm (e.g. Larsson's algorithm). In ampycloud, setting B to 30% equates to only using 30% of the data within a slice. Is this correct? If not, could you please add a few words to clarify?

Line 164-165: "as commonly done by other cloud height algorithms" Are there any references to this?

Line 209: why is alpha_s chosen as 0.2?

Line 220: perhaps refer again to the over-slicing as the top layer of figure 1 to add clarity
Line 229: Please give a reason for choosing epsilon as 10%
Line 245: how is fb chosen?

Line: 250 how is Lfrac chosen?

Line 330: Could you please give more information on how a human observer collects/validates data for his METAR: they derive a cloud layer height and amount based on the clouds in an area above and in the viscinity of the aerodrome. But do they also use raw ceilometer cloud hits? Do they use the results of another algorithm? Are they trained to more or less estimate cloud heights without ceilometer information?

Line 332: how were the cases selected? Did you only select cases where human observers reported something operationally significant? Or did you select cases where either a human or the algorithm detected something important? Or did you select cases where both the algorithm and the human observers agreed that something of operational significance was occurred? Or was there another criteria used? Please specify as this could have consequences on the number of misses and false alarms. Also, can you say if using a larger sample (for example by using the whole 5 years) would affect the statistics?

Line 336: What is meant with operationally significant cloud events: are these ceilings? Low clouds? Low ceilings? Situations where at least 2 layers are present with at least a SCT in the second layer? And since these represent such a small sample compared to the whole 5 year dataset, does that mean that ampycloud is designed to be most accurate for operationally significant events at the expense of other events? Please clarify this.

Line 338-344: what is meant by a density of a cloud? Is it the number of oktas?

Line 346: lines 333-336 seem to suggest that the 2128 cases are the high impact cases. In line 346 a further selection is made? Could you please clarify this?

Line 367-369: "Understanding the exact behavioral differences between ampycloud and other algorithms with similar purpose would require a dedicated comparison of their respective performances against a reference dataset, which is outside the scope of this article." Please provide more detail as to why such an analysis is not done. It is true that algorithms from manufacturers are kept secret, but in the case of Vaisala it is possible (against payment) to have the outcome of their algorithm in terms of cloud layer heights and cloud amount. Could you explain why a comparison of this output against ampycloud is not done? Furthermore, what is meant with a reference data set in this case?

Line 415-417: Should this be in the results section?

Line 419-429: In the method section?

Figure B10: low visibility procedures should be introduced

Figure B12: information about VV being decided by a separate algorithm should be in the main text in the introduction or methodology section, together as to why meteoswiss/the authors decided this that way. Also, as VV is not within ampycloud, it should be emphasized that ampycloud is only partly compliant with ICAO's rules for cloud reporting.

**Minor comments**

Please use 'inputted' instead of 'fed'

Lines 94-95: starting the sentences with 'It' may make the sentences more readable

Lines 114-115: Better to write: "The longer the time interval or the larger the wind speed, the better the spatial representativity of the dataset, but the worse is the view of the current state of the sky"

Line 127: use 'apply' instead of 'run'

Line 353-355: "But it cannot guarantee that the "correct" cloud layers are being identified when they have the correct density." Perhaps better to say that detecting the right number of oktas for a cloud layer cannot guarantee that the cloud layer has the correct height?

Line 390: small amount of information implies that you are talking about a small number of cloud hits, which is not the case. In this sentence, please consider saying that it relies on cloud hits only.

Line 399: use the word "detect" instead of "see"

Line 489: Please define "scientific stability"

---

## Author Response (AR1)

We thank the anonymous reviewers for their extensive and constructive feedback, which allowed us to improve the manuscript in several parts. Our detailed answer to each point raised is included below. All additions/modifications in the article have been underlined in the draft. Some of the comments have led us to update the ampycloud Python code, such that the article now corresponds to v2.0.0 (currently available in the `develop` branch of the GitHub repository, that will be formally released upon acceptance of this article for publication). We note however that the scientific performance of version 2.0.0 of the code remains identical to that of v1.0.0 (the changes are essentially related to naming conventions).

The authors
May 2024

**------------------------------------------------------------**
**Reviewer 1**
**General comments**

1) The preprint entitled "ampycloud: an algorithm to characterize cloud layers above aerodromes using ceilometer measurements" has been reviewed for possible publication in Atmos. Meas. Tech. The authors developed a new algorithm and associated Python package claiming it is suitable for the fully automated reporting of cloud information relevant for Meteorological Aerodrome Report (METAR) at Swiss civil aerodromes. The open source strategy with the Python package and its online documentation are positive, but the potential user group is likely limited to other weather services.

The manuscript is clearly organised, although individual paragraphs and figures belong in other sections or in the appendix. The diagnostic diagrams are really nice, although they do require a certain amount of background knowledge. The writing style is well understandable, even if some issues could have been explained better with more appropriate wording. This would also avoid unnecessary repetition of the same content both in the main sections and in the figure captions. Spelling and grammar are appropriate but need improvement and British English should be used consequently. Occasionally, the authors become entangled in contradictions and misunderstandings are inevitable for less knowledgeable readers. In a few places there is no corresponding reference, e.g. to the CL31 manual from Vaisala (2009).

Reply: We are glad to read that the reviewer supports the open-source strategy. The reference to the CL31 ceilometer has been added. We agree that it is unusual to have specific information provided only within figure captions. In the present case however, we are strongly convinced that doing so significantly enhances the readability of the article, in particular for the figures in the Appendix that would otherwise require a tedious back-and-forth between the text and the diagrams. We note that AMT accepts "all standards forms of English in order to retain the author's voice" (https://www.atmospheric-measurement-techniques.net/submission.html#english), and we prefer to use US-English throughout.

2) Essentially, it is a matter of assigning the cloud hits originating from n ceilometers, which are derived from the actually measured backscatter signal, to cloud clusters, summarising these as far as possible into cloud layers and finally weighing up which of these are relevant for reporting, considering all the requirements and rules of the ICAO (2018). The authors use interesting approaches applying agglomerative clustering method and Gaussian mixture model, but it seems like they take a sledgehammer to crack a nut. In contrast, some key requirements of the ICAO (2018) are not met like reporting "NSC" (nil significant cloud) or vertical visibility (VV) with altitude value (see specific comments).

Reply: Ceilometer data alone are not sufficient to formally identify an NSC situation. NSC should be used if there are no clouds of operational significance and no restriction on vertical visibility and the abbreviation 'CAVOK' is not appropriate (see EU373/2017 AMC1 MET.TR.205(e)(1)(d)). Note that clouds of operational significance include the convective clouds CB and TCU (see EU373/2017, Annex I – Part-Definitions). At MeteoSwiss, convective clouds are detected by a separate algorithm based on lightning and weather radar data, while CAVOK is determined from a combination of horizontal visibility, output of ampycloud and present weather algorithms. Consequently, the decision on whether the situation is NCD or NSC cannot be done by ampycloud alone. Nonetheless, we do agree with the comment that ampycloud should not be solely issuing NCD. As of v2.0.0 of the code, assembled as a result of this review process, ampycloud will now issue NSC and NCD codes based on the ceilometer data at hand. The limited ability of ampycloud to issue "absolute" NSC/NCD codes is now discussed in Sec. 2.2.

Regarding the VV aspect, the text has been reformulated to make it clear that ampycloud is one among many AUTO METAR algorithms within the larger software SMART. The decision to issue a VV code over a cloud base is handled by a distinct algorithm that relies on additional input information. Specifically, we had to introduce additional criteria to improve the distinction between VV and low ceiling cases based on present weather and horizontal visibility information – the main rationale beeing that if horizontal visibility is reduced due to snow or fog, we are more likely to be in a VV situation than not. Given the complexity of the decision tree of that separate algorithm, it has deliberately been kept separated from the core cloud detection algorithm (ampycloud), which focuses solely on determining the sky coverage fraction and base height of cloud layers using ceilometer data. Of course, the output of the VV algorithm overrides the one of ampycloud during a VV situation.

Regarding the apparent complexity of ampycloud over other existing algorithms, we believe that this impression may be driven (in large) by our efforts to ensure that our work is fully reproducible. Other existing algorithms may also not be as simple as they appear on the surface: after all, the ASOS user manual itself states that "describing exactly how the algorithms work is quite complicated, so what follows is a simplified explanation." The CL31 user's guide describes the sky condition algorithm only at a high level, which is understandable as the software is proprietary. Nonetheless, its description already shows that multiple processing steps are necessary (e.g. "an aggregation steps follows the initial clustering"). Finally, we also received an informal document describing the implementation of the algorithm of Larsson and Esbjörn (1995) at Skeyes (courtesy of F. Chatterjee). The description reveals a rather complex algorithm procedure that includes both slicing-like and aggregation steps, among others. Finally, we note that we also have been unable to locate the Larsson and Esbjörn (1995) reference quoted by Wauben (2002) as the source for the KNMI algorithm, and this even by contacting the Swedish Meteorological and Hydrological Institute (SHMI), that could not tell us what this reference corresponds to.

3) The quality of any similar cloud algorithm depends strongly on the performance of the ceilometer type used. Various ceilometers from the same or different manufacturers derive from their attenuated backscatter profile different cloud and VV hits at a given time and each device has its own strengths and weaknesses (e.g. Illingworth et al., 2019). ampycloud is designed and tuned for CL31 and needs to be adapted or even extended if you replace the ceilometer. In the specific comments, I try to explain the limitations of the CL31 with regard to its use in aviation based on personal experience.

Reply: ampycloud has been designed to process cloud and VV hits irrespective of the systems from which they originate. Throughout the article, we clearly state that the handling of VV hits and the time interval Delta t are for the user to decide/set. From this perspective, there is no specific "tuning" of the algorithm in favor of the CL31, even though it remains very true that we have not formally tested the algorithm with other ceilometers. The fact that ampycloud depends strongly on the performance of a given ceilometer to detect clouds in the first place is clearly discussed in Sec. 4, where we have added a reference to Illingworth et al. (2019). ampycloud cannot (and should not) compensate for a poor ceilometer performance: it merely focuses on finding the best clusters/layers given a set of cloud hits.

4) From the results presented, I cannot recognise any superiority of ampycloud over existing similar cloud algorithms, except that the implementation here is significantly more complex. Furthermore, its assessment based on observations is not convincing. Contrary to what the authors call an improvement of the algorithm, it would be an actual innovation with potential to be published, if a) the integration time has been made dependent on the horizontal wind speed or b) the measured backscatter signal has been used to derive cloud hits and VV hits by yourself to get rid of the manufacturer black box algorithms. In my opinion, this manuscript is not suitable for publication in its current form.

Reply: It certainly was not our intention to claim that ampycloud is superior to other existing cloud algorithms. The text has been reviewed to ensure that there is no possible misunderstanding about this (see also our replies to 6.3, 6.6 and 6.7 below).

ampycloud was born as a necessity, as MeteoSwiss moves towards the fully automated generation of METARs. As already mentioned at point 2, it is the lack of detailed documentation and open-source codes that led us to develop our own algorithm, and that motivates the submission of this article today. ampycloud is certainly not the final answer to the problem, but it has nonetheless been found to be fit-for-purpose by MeteoSwiss and the Swiss Civil Aviation Authority. By releasing the code and publishing this article describing its scientific inner-workings, we very much hope to stimulate intercomparisons and further developments of this algorithm, and possibly others as well.

As already mentioned at point 3, ampycloud cannot and should not compensate for a poor ceilometer performance, but only focus on finding the best clusters given a set of cloud hits. ampycloud never aimed at removing the manufacturer black-box algorithms that derive the VV or cloud base hits, but rather to remove the black-box sky condition algorithms.

**Specific comments**

5) My recommendation for the title is:

"ampycloud: an algorithm for determining the amount and base altitude of cloud layers over aerodromes using ceilometer data"

On the one hand, there are numerous macro- and microphysical cloud properties (e.g. Gao et al., 2014; Luebke et al., 2022), but ampycloud is only able to determine cloud amount and cloud base altitude based on the data of n ceilometers. On the other hand, the quantity actually measured by ceilometers is the backscatter signal (attenuated backscatter), while the cloud or VV hits are derived by application of manufacturer-specific algorithms due to the lack of a quantitative definition of a cloud (section 2.1.1 of WMO, 2017).

Reply: The title has been adjusted to provide a more explicit description of the algorithm's goal, and now reads:

"ampycloud: an open-source algorithm to determine cloud base heights and sky coverage fractions from ceilometer data"

6) ampycloud is a strongly scientific rather than application-orientated implementation. There is no need to determine cloud amount and cloud base altitude with best accuracy, considering that for the METAR/SPECI the former is classified into FEW, SCT, BKN, OVC and the latter is rounded down to 100 ft increments below 10,000 ft altitude. The current version of ampycloud is only suitable to a limited extent for the complete, operational determination and reporting of cloud and VV information in METAR/SPECI or MET REPORT/SPECIAL syntax, i.e. major revisions are required:

Reply: As discussed in Sec. 2.1, we are of the opinion that strongly scientific/physics-based algorithms can prove to be very beneficial also in application-oriented environments. In particular, we are convinced that the former does not preclude the latter (provided, of course, that the implementation of the algorithm recognizes the restrictions imposed by an operational environment). ampycloud is one element within a larger set of algorithms and tools used to produce AUTO METARs at MeteoSwiss. The manuscript has been revised to remove any possible source of confusion regarding this aspect.

6-1)
• NSC is currently not considered but required by section 4.5.4 of ICAO (2018). Your Fig. B7 is just such a case where you have to report "NSC" in the METAR because there are clouds (above MSA) but they have no operational significance.

Reply: As of v2.0.0 of the code, the ampycloud package will report NSC and NCD codes based on the ceilometer data available. This is now correctly illustrated in Fig. B7 (we had initially disabled the MSA for this exemple, which we agree was confusing). See also our reply to point 2.

6-2)
• Height/altitude values for VV can currently neither be determined nor reported although it is a requirement according to sections 4.5.4 and 4.6.5.1 of ICAO (2018). The CL31 provides VV hits and I wonder why the authors argue that another algorithm should handle it. Your data in Fig. B12 could also mean a so-called VV case, which

leads to the possible result "VV001" in the METAR.

Reply: We do not argue that VV hits should absolutely be handled separately, only that this is currently the case at MeteoSwiss. The article text has been adjusted to remove any source of possible misunderstanding on this front. See more details in the answer to point 2.

6-3)
• Your line of argument for using a shorter integration time of Delta t = 15 min is flawed. Such a short time period does not necessarily require a weighting of the latest cloud hits in order to better react to rapid changes in the cloud field, but e.g. for 30 min it makes perfect sense. METARs are representative for 8 km (even 12 km for convective clouds) radius around the airport reference point and the reporting time is 30 min. Nadolski (1998) pointed out that "… a 30-minute time period provided an optimally representative and responsive observation similar to that depicted by an observer". Do you have observational evidence that your choice is better ? Depending on the wind direction, the ceilometer detects clouds in the downstream half-space, i.e. only those clouds that have passed over the device. The crucial problem with a shorter Delta t is the loss of information, which leads to less agreement with the observer who is looking at the entire half-space.

Reply: It was not our intention to imply that all algorithms applying some weighting schemes are illogical. The corresponding sentences in the third paragraph of Sec. 2.5 have been adjusted to avoid all misunderstanding on that front. Regarding the use of the time interval, it is entirely up to the user to decide how far back to look (which can also depends on stakeholder requirements). MeteoSwiss has been using 15 min for numerous years.

We note that the 8 km spatial representativity of METAR only applies to present weather elements that are not in its vicinity e.g. VCFG, VCSH, VCTS (approximately 8-16 km, see EU373/2017 AMC MET.TR.205(d)(3)). Concerning the spatial representativity of clouds, regulation states that "Cloud observations for METAR and SPECI should be representative of the aerodrome and its vicinity" (see EU373/2017 AMC1 MET.TR.210(e)(b)). It is understood that "vicinity" in the cloud context should not be interpreted as approximately 8-16 km as in the present weather context, otherwise we would miss the clouds within 8 km. In addition, the regulation does not seem to specify any difference between the vicinity that should be used for cloud height and amount, which is reached through longer or shorter temporal integration times of ceilometer data, and the vicinity that should be used for convective clouds CB/TCU, which spans from 16 km up to 50-60 km depending on the European country. While there are still ongoing discussions in the MetAlliance ET-AUTO OBS and ET-AMR on how to interpret the regulation, users and stakeholders can essentially choose whatever value they like, as long as it is apporved by the Competent Aviation Authority.

6-4)
• I wonder why the authors explain ampycloud using Fig. 1, where obviously an integration time of Delta t = 20 min (1200 s) was used, but they only speak of Delta t = 15 min (900 s) both in the main text and in the figure captions of the appendix. This is an inconsistency and leads to misunderstandings. Why did you switch from 20 minutes to 15?

Reply: Fig. 1 was updated to use 15 min and avoid confusion. However, we do note once again that the decision to analyze ceilometer hits over 15, 20 or 30 min is entirely up to the user — ampycloud will simply characterize the data that one feeds it.

6-5-a)
• Your slicing step applies the Manhattan metric as a distance function and average linkage as a fusion method. Rescaling the time axis is a clever hack to reduce the 2-D problem to 1-D, but why are you still using a 2-D clustering method that is computationally less efficient and slower? The original (historical) implementation of the ampycloud algorithm is no reason to deny further developments.

Reply: We fully agree, and this point is now mentioned explicitly in Sec. 4. We would like to remind again that our intention is not to sell ampycloud as the final solution, but rather as an algorithm that can (and will) be improved iteratively over time. This specific point will definitely be considered in one of the next software releases.

6-5-b)
From your initial slicing step, which is similar to the binning and clustering steps of the ASOS (Nadolski, 1998) you have generated two new problems: 1. To combine two cloud clusters that were incorrectly assigned to different slices but actually belong to each other you use an alternative distance function (Euclidian metric) and fusion method (single linkage). In the latter outliers often give rise to chain effects ("single-link effect") and tiny clusters that consist of only a few elements. 2. The problem that two independent cloud clusters were incorrectly assigned to the same slice you try to solve using Gaussian mixture models. Wouldn't it be more plausible to use Gaussian mixture models to combine cloud clusters to form final cloud layers? Models with lower BIC are generally preferred but a lower BIC does not necessarily indicate one model is better than another. Have you applied other distance functions and fusion methods for the slicing and grouping steps to be sure what is the ideal choice? Is the Gaussian distribution really suitable for cloud hits in the layering step? I doubt these questions have been adequately answered.

Reply: We have tried a variety of approaches when developing ampycloud, and the version presented in this article is the most adequate solution we could identify so far (both in terms of absolute performance, and as a trade-off between complexity and robustness). As described in the text, the key is to identify a simple-and-robust approach, and correct its specific (well understood) shortcomings. It is evident that many alternatives exist, and we do not claim that our solution is "the ideal choice" — simply that it has been found to be fit-for-purpose. The many possibilities for improvement, including for the layering step, are discussed in Sec. 4.

6-6)
• The lack of comparison with the results of at least one similar cloud algorithm already in operation. The authors claim in section 4 that this "… is outside the scope of this article.", but they need to show the added value of ampycloud to justify the higher computational effort and complexity.

Reply: It is certainly not our intention to claim that ampycloud is superior to other existing cloud algorithms — only that we have found it to be fit-for-purpose. The text has been adjusted to ensure that there is no possible confusion regarding our intentions. As explained at point 2, the closed-source nature of the other cloud algorithms very much prevents us from performing a detailed comparison (that we fully agree would be very interesting). We do note, however, that members of the Met Alliance ET-AUTO OBS (http://www.met-alliance.com/) are currently discussing the possibility to organize one such intercomparison of cloud algorithms.

6-7)
• In section 3.4 the authors mention to use "a reference dataset of 2128 cases extracted over a 5 year period (2018-2022) from LSGG METARs" for the statistical evaluation and assessment of ampycloud. Even if SPECIs are not considered, a single year comprises 17520 METARs (twice an hour for 365 days). On the other hand, 2128 cases correspond to about 44 days. In relation to the 5-year period, it is only slightly more than 2.4 %. From a statistical point of view, this is definitely insufficient, your results are therefore not significant and the pre-selection of cases harbours the risk of concealing serious shortcomings in your algorithm.

Reply: Our selection of cases was made with the explicit goal of assessing the performance of ampycloud under challenging conditions. For instance, the ratio of cases with a ceiling (as reported by the observer) in our sample of 2128 cases is 71.5%. Had we blindly taken all METARs, this number would have decreased to 36%, as a result of the inclusion of a significant number of "easy" cases (NCD/NSC, single cloud layer, etc ...). Clearly, assessing ampycloud blindly over all possible METARs would lead to improved performances solely because one is "diluting" the hard cases (see for example the very well-known Tornado Finley example: https://www.cawcr.gov.au/projects/verification/Finley/Finley_Tornados.html). We voluntarily created a more balanced dataset containing a better proportion of complex and easy cases, which avoided using accuracy scores designed for unbalanced dataset, see for example the (less interpretable) scores for rare events at https://www.cawcr.gov.au/projects/verification/#Methods_for_rare_events. We thus remain of the opinion that our sample of 2128 cases does present a more interesting and scientifically appropriate assessment of the accuracy of ampycloud to be discussed in the article.

6-8)
• There is neither a controversial discussion nor at least a citation to the strengths and weaknesses of the observer, who is not the "ground truth" in all situations. This must be considered when discussing the results of ampycloud with regard to a possible under- or overestimation.

Reply: We thank the reviewer for this comment. We extended the discussion in Section 3.4 to make the reader aware of the comparison challenges between automatic and human observations.

6-9)
• The authors only present results for ideal input data, but this does not always apply to fully automatic operation. Please provide more detailed insights into the exception handling. How does your algorithm react if …
o there is a "complete failure", e.g. due to power or network failure, and no input data is (temporarily) available?
o there is a failure of individual or multiple ceilometers (e.g. due to maintenance)?
o a single ceilometer measures but transmits an error code?

Reply: Ampycloud treats the ceilometer data it receives and automatically accounts for a reduced number of time stamps in the computation of cloud amount. This covers the cases when ceilometer data have gaps due to technical issues. At Geneva airport threre are four ceilometers, so if one has a technical failure or an error code, we continue to produce AUTO METARs with the other three. If there is a complete failure, ampycloud will not be called by the SMART software (i.e. the exception handling is done before calling ampycloud). It is the responsibility of the user to decide below which threshold it is not worth to call ampycloud. This aspect is now made clear in Sec. 4.

6-10)
• Some important information about the CL31 is missing, e.g. the output interval of 15 s (4 hits per minute), which can only be inferred from the max. hits per layer in the diagnostic diagrams with some difficulty. The Vaisala CL31 algorithm derives from the actual measured backscatter signal either up to three cloud hits in different heights or a value for vertical optical range (VOR) above ground level at any given time. Similar to determining cloud base altitude VV hits should be used to derive an altitude value for VV. By definition, VOR is the height above the ground up to which the extinction coefficient must be integrated in order to reach a certain threshold value, usually around 3 (Werner et al., 2005). VOR is always higher than all cloud hits detected at a given time, because a well-performing ceilometer can always "see" a little more optically thick clouds and even through optically thin clouds. However, Vaisala obviously uses a significantly smaller threshold value in order to have an alternative definition for the derivation of cloud hits. They call it a VV hit, but it's basically a cloud hit. Figures B1 and B2 show very nicely that a) cloud hits and VV hits merge almost seamlessly into one another and b) ampycloud would be completely wrong if the VV hits were not considered cloud hits. The crucial problem is that these two quantities are completely different things from a physical/meteorological point of view.

Reply: The reference to the CL31 user manual has been added, together with the measurement rate. ampycloud is designed in such a way that it is agnostic of the origin of the cloud/VV hit — it only seeks to characterize the resulting hit distribution. We do not think that the details of the CBH/VV detection algorithm fall within the scope of this article, in particular because they vary between systems, and it would not be fair to focus on one system in particular. As mentioned clearly in Sec. 2.3: "It is up to the user to decide if and when VV hits should be provided to ampycloud, and whether they need to be pre-processed in any way before doing so". We also do not agree with the statement that the "CL31 VV hits are basically cloud base hits". When the output of ampycloud is overridden by the vertical visibility algorithm, we noticed that VV hits are systematically higher than cloud base hits, which is more in line with human observations. As mentioned at point 2, it is outside of the scope of this article to present such vertical visibility algorithm, which also depends on horizontal visibility and present weather data.

**In the following there are specific comments to certain lines or figures of the preprint:**

7) ll 24 MET REPORT/SPECIAL is not representative of the runway, but of the threshold. Each runway has 2 thresholds. Unfortunately, the MET REPORT/SPECIAL syntax is e.g. "... CLD RWY 05 OVC 800FT ...", which can lead to confusion.

Reply: Agreed. We rephrased the sentence in text.

8) ll 84 There are also parameters that have only computational or tuning relevance, such as your OVC/FEW thresholds or alpha_s in Tab. 2, which are arbitrarily chosen.

Reply: As mentionned in the text, "ampycloud was developed with these six requirements in mind". We do not claim that all the parameter values in Table 1 have a strict physical justification.

9) Fig. 1 Figure 1 should be presented in section 2.4 if the abbreviations and parameters have already been introduced. The figure caption of Fig. 1 is very long. I prefer to describe the method and general information in the main text with reference to this figure. All figures in Appendix B and Fig. 1 should then be self-explanatory.

Reply: We agree that it is uncommon to have such an extensive caption, and would typically prefer shorter ones as well. In the present case, however, we believe that the (necessary) complexity of the ampycloud diagnostic diagram, together with its repeated use throughout the article, does warrant (and in fact require) the legend of Fig. 1 to be exhaustive. Doing so provides the reader with direct access to the relevant information necessary to understand/identify every element of the diagram. Having to refer to the text would imply having to read through Secs. 2.3 to 2.5 to understand the content of the diagram, which we are convinced would be highly detrimental to the readability/usability of the article. As suggested, this figure is now introduced at the end of Sec. 2.4.

10) ll 120 – As you mention in relation to "nnn" in the caption of Fig. 1, a ceilometer does not provide vertical values relative to mean sea level (altitude = elevation + height), but it gives the height above ground level. The elevation corresponding to a specific location must be explicitly added to the settings.

Reply: We thank the reviewer for bringing this point to our attention. The use of the correct terminology (namely, "height") has been implemented throughout the article (incl. the figure) and the code. We note that the MSAs quoted in the figures were indeed specified in ft above aerodrome level (which is consistent with https://www.meteoschweiz.admin.ch/service-und-publikationen/publikationen/verschiedenes/2023/flugwetterinformationen-in-der-schweiz.html, and ensures for a direct compatibility with ceilometer hits). This has now been made explicit everywhere.

11) lls 127 – 128 and 134 – 135
"Data are pooled together before the analysis, independently of the spatial distribution of the ceilometers." and "The specification of hcid for every hit is necessary to compute a correct estimation of …" are contradictory statements.

Reply: The sentences have been brought together and the first one re-worded to lift the apparent contradiction, as follows:

"Data from different ceilometers are analyzed together, independently of the spatial distribution of the instruments on the ground. The specification of $h_{\rm cid}$ for every hit ensures that a correct estimation of the sky covering fraction (cloud amount) can be made, under the assumption that cloud layers have a unique base per time step per ceilometer line-of-sight."

12) ll 140 What is meant with "signal range"? Is it the maximum range of detection?

Reply: Yes.

13) Fig. 2 – Figure 2 should not be shown in the main sections, but at most in the appendix, as it is essentially just a repetition of the results from section 5 of Wauben (2002). The added value of n > 1 ceilometers for distinguishing BKN and OVC is rather limited, as both mean a ceiling in the end.

Reply: It is certainly true that Wauben (2002) does discuss the question of multiple ceilometers, but we do not think that Fig. 2 (now Fig. 1) is "just a repetion" of that work. Wauben (2002) reports experimental evidence, whereas our figure provides a theoretical view on the matter. We do promptly refer to Wauben (2002) in the text, so that it is evident for the reader that we are not the first to discuss/make this point. Given that no such figure is presented by Wauben (2002), we think that its presence in the core of our article remains justified to support the statement regarding the impact of multiple ceilometers.

We do agree that the ability to better distinguish between BKN and OVC would have little operational impact, and we have updated the text as follows:

"The ability to better distinguish between OVC and BKN layers may not be a crucial benefit in itself, given that both categories imply the existence of an (operationally-relevant) ``ceiling''. However, using multiple ceilometers can also contribute to reducing cases of gross over-/underestimation of sky coverage fractions in slow-moving conditions (i.e. low-wind with quasi-stationary clouds), provided that the horizontal separation between ceilometers is larger than the applicable cloud characteristic scale length \citep{Slobodda2015,Denby2022}."

14) ll 160 – Meanwhile Vaisala provides an "airplane filter" and maybe it can even be utilized by CL31 with a firmware upgrade.

Reply: An explanatory footnote has been added to this sentence.

15) lls 262 – 263 – The triangles and rectangles in the blue slice in Fig. 1 reveal that alpha_s * (hmax – hmin) ~ 0.2 * (2,100 ft – 950 ft) = 230 ft. The distance between the two sub-layers is definitely larger than 230 ft. So why are the two combined during the slicing step?

Reply: The vertical rescaling for the slicing is performed over all the hits, i.e. hmin = 750 ft & hmax = 6250 ft, such that alpha_s * (hmax – hmin) = ~1100 ft, as indicated in the secondary y axis to the right of the diagram. This explains why the triangle and squares are not separated into two distinct slices.

16) lls 350 – 351 – Or the differences are due to the observer's limited ability to distinguish between slightly more or less than 50 % cloud cover.

Reply: Agreed. The text has been adjusted accordingly.

17) ll 357 – Lower cloud bases occur almost twelve times (24.6 %) as often as higher ones (2.1 %). Therefore, it is not a tendency but systematic. This is potentially problematic when applying the SPECI criterion in section 2.3.2 f) of ICAO (2018).

Reply: The sentence has been reworded as follows:

"The majority of the remaining cases are comprised of situations when \textsf{ampycloud} provides somewhat lower cloud heights than the METARs." The systematic nature of this behavior is already explained in the next sentence of the article. This is not problematic in terms of the SPECI criterion 2.3.2 f), in the sense that ampycloud does not "wrongly underestimates" the height of cloud layers – rather, it merely takes a conservative approach when characterizing a cloud layer with time-varying height (an approach than can be adjusted by users by means of the beta_t parameter intorduced in Sec. 2.4).

18) ll 360 – Or your Delta_t is too short and you miss some clouds that passed over CL31 earlier.

Reply: The sentence has been amended as follows: "Cases in the bottom row (6.6% of the total) correspond to situations where the cloud layer is being missed entirely by the ceilometers over the duration of the time interval Delta t." We note that increasing Delta_t at Geneva airport can only partially reduce this issue as there are often cumulus (humilis and/or mediocris, that are not operationally relevant) that remain stationary over the Jura mountain ridge (12 km away from the airport). For example, out of the 5 examples in Appendix B where the ceilometers miss entirely a cloud layer (B4, B6, B7, B8, B10), none would have seen the missed layer had the time interval been increased to 1800s. This point has been added to Sec. 4.

19) Fig. B4 Perhaps it is the same cloud layer that has passed over at least 2 ceilometers with a time delay. Then the two cloud clusters (green and red) should be combined into a single cloud layer. This would be more consistent with the 2nd cloud group reported by the observer.

Reply: The merging of the green and red layer would not suffice to obtain a SCT layer, there simply is not enough hits between them (both are only 1 okta). If the user still would like to merge these cases, ampycloud provides the parameters MIN_SEP_VALS and MIN_SEP_LIMS. These allow to set a minimum separation between layers at a given height for re-merging.

20) Fig. B5 The cloud hits around 9,100 ft and 9,500 ft belong definitely to a single cloud layer, as it is a similar situation as for the red slice in Fig. B6. All hits together would likely result in SCT.

Reply: As visible in the middle column (corresponding to groups), having all the hits at 9000 ft and 9500 ft together would still only result in a FEW. Reference to this element of the plot has been improved in the legend. As mentioned at point 19, if the user requires so, the default ampycloud parameters can be changed to improve the handling of these cases.

21) Fig. B6 The cloud hit distribution and your slicing step clearly speak in favour of 3 cloud layers similar to the observer-based METAR meaning that the grouping and layering steps fail.

Reply: The grouping and layering steps do not fail, in the sense that they are peforming as as expected. Whether the outcome is correct is a different matter, and the caption certainly does not claim explicit success in this case. For the record, we did conduct a few months back an informal expriment asking ceilometer-expert colleagues whether this hit distribution contains 1, 2, or 3 layers, at first without giving them any a priori information (hits were not colored, and the METAR was not given). Hardly anyone saw 3 layers in the distribution ... that is until we show the hit coloring based on the slicing. This is precisly why this figure has been included in the article.

22) Fig. B10 Perhaps your Delta_t is too short and you therefore cannot reproduce the clouds at around 3,500 feet.

Reply: See our reply to 18). The same clarification was added to this caption and those of the Figs. B4, B6, B7 and B8.

**Technical corrections**

23) lls 1, 15, 45, … Either "sky coverage" or "cloud amount" but use a consistent wording.

Reply: Sky coverage is now being used throughout the article.

24) ll 28 – I would prefer "international" rather than "civil" airports, as regional airports as a different category are not limited to goods and trade.

Reply: In Switzerland, MeteoSwiss is also involved in the issuance of METARs at military aerodromes, with similar–yet–formally distinct software tools. At the same time, SMART is being used both at international and regional civil aerodromes. We thus prefer to use the term "civil", which correctly identifies the aerodromes which we refer to in the article.

25) ll 29 – "in addition" instead of "an additional"

Reply: "additional" has been removed from the sentence.

26) ll 59 – "cloud layers" without 2nd "s"

Reply: Corrected.

27) ll 61 – "its performance" singular is enough

Reply: Corrected.

28) lls 66 – 71 – This paragraph should be moved to section 2 or even section 3.

Reply: Agreed – the paragraph was moved to Sec. 3.1.

29) lls 98 – 99 – Move these two sentences and Fig. 1 to section 2.5 and shortly refer to it here avoiding repetition of the same content.

Reply: Agreed – Fig. 1 (now Fig. 2) is introduced at the very end of Section 2.4.

30) ll 111 – Use "cloud and VV hits" instead of "ceilometer hits".

Reply: Adjusted throughout the article.

31) Fig. 1 – Delta_t = 20 min should be the same as for figures in Appendix B (15 min)
– omit sup-/subscripts and explanation of OVC/FEW thresholds as these
  information can also be found in Tab. 2.
– which airport is shown? (MSA: 8,000 ft refers to LSZH but nceilos: 4 to LSGG)
– max. hits per layer: 160 is incorrect
  nceilos * hits/min * Delta_t = 4 * 4 * 20 = 320 (doesn't fit) but = 2 * 4 * 20 = 160 (fits)

Reply: The data presented in Figure 1 are artificial. They are comprised of mock observations designed to illustrate the complete behavior of the ampycloud algorithm as "cleanly" as possible. The artifical nature of the data is written exlicitely in the caption as well as in the figure itself. As such, the data do not belong to any specific airport, and the MSA is purposely chosen to not be relevant. We have changed the lookback time to 900s and increased the data rate (from one observation every 30s to one every 15s) for better consistency with the other exemples –– and this even though the lookback time is a choice to be made freely by the user depending on their preferences. Sup-/subscripts explanations have been removed.

32) ll 117 – Use "cloud hit" instead of "ceilometer hit".

Reply: Adjusted throughout.

33) ll 120 – "derived cloud base height above ground level" or "derived cloud base altitude above mean sea level" but no mixture and not including "measured"

Reply: Corrected ("height" is now used throughout).

34) ll 124 – "backscatter profiles"

Reply: Corrected.

35) Tab. 1 Maybe add the CL31 output interval of 15 s, the elevation, MSA, etc.

Reply: Done.

36) ll 148 – use "time interval", "time increment" or "integration time" but not "length"
– "can be bundled" instead of "can be pooled"

Reply: Corrected.

37) ll 151 – "… time difference relative to the latest."

Reply: Corrected.

38) lls 157 – 159 "Assuming max. hits per layer is 160, then we use a threshold value of 99.375 % for OVC and 1.875 % for FEW, accepting 1 hole for the former and requiring 4 hits for the latter." Your wording is far too cumbersome. Why do you apply an implicit dependency on max. hits per layer instead of defining fixed percentages?

Reply: We cannot find this sentence anywhere in the article. In particular, lls 157 – 159 read: "Theta_0 is the maximum (absolute) number of cloud hits for a given slice/group/layer to still be considered to be NCD by ampycloud; Theta_8, on the other hand, is the maximum number of "holes" (i.e. non-detection of clouds) for a given slice/group/layer to be considered to be OVC by ampycloud."

We favor absolute instead of relative thresholds as they are much better suited to accurately handle small numbers of "misses" (also with very few ceilometers). Absolute numbers are also much easier to understand as a parameter from the user perspective: when using relative ones in an earlier phase, we found ourselves constantly converting the percentages back to absolute counts.

39) lls 178 – 179 – "This distance threshold is Delta_hl = 250 ft between 0 – 10,000 ft (1,000 ft above 10,000 ft)." Your wording is rather cryptic.

Reply: As for 38), we cannot find this sentence anywhere in the submitted article.

40) lls 186 – 187 – such repetitions should be avoided

Reply: We are not seeing any kind of repetitions in the lines indicated.

41) Tab. 2 – "behaviour" in the figure caption
– table too long and covers page number
– add the meaning of parameters in a separate column and omit Python variable

Reply: We prefer to use US-english in this article. The Table length has been fixed. We prefer not to include the meaning of the parameters in the Table, to avoid un- necessary repetition with the main text. For some parameters, explaining their meaning is also difficult without context. Instead, we have added references to the Sections where the different elements are introduced. Finally, we prefer to keep the link to the Python variable to ensure that the article can be used efficiently as a scientific complement to the Python package.

42) ll 206 – misleading footnote (could be power 3)

Reply: Corrected.

43) lls 222 – 223 – "By (our) definition, slices i and j overlap if one of the following two conditions is true/met:"

Reply: Corrected.

44) ll 227 – use "For example, the two top slices in Fig. 1 (green and red) overlap, …"

Reply: Corrected.

45) ll 241 – use "the latter value" instead of "this value"

Reply: Corrected.

46) ll 250 – A reference to Tab. 2 is missing for Lfrac and Lit

Reply: Table 2 has been adjusted with Lfrac and Lit now listed under the Grouping step.

47) ll 271 – "overestimation" instead of "overestimate"

Reply: Corrected.

48) ll 276 – use precise wording to avoid "In other words, ..."

Reply: "In other words" has been replaced by "Essentially".

49) ll 277 – use "assumes" and "number" instead of "will deem" and "amount", respectively

Reply: "assumes" adjusted as suggested, "score" used instead of "value".

50) ll 300 – Avoid text extending beyond the edge of the page.

Reply: Corrected.

51) ll 304 – the correct reference is "(ampycloud, 2024a)

Reply: Corrected.

52) ll 334 – use "5-year period"

Reply: Corrected.

53) ll 348 – use "The comparison of Fig. 3 and Fig. 4 shows, that false alarms …"

Reply: Corrected.

54) ll 355 – "height/altitude of the cloud layer with the highest degree of coverage" instead of "densest cloud layer"

Reply: Corrected.

55) ll 357 – "than" instead of "that"

Reply: Corrected.

56) ll 359 – consistently use either "cloud base height" or "cloud base altitude"

Reply: Agreed and done ("height" used throughout).

57) Fig. 5 maybe use "hft" (hecto feet) instead of "100 ft"

Reply: We prefer to use 100 ft, as the unit hft is not standard/less explicit, and would be more confusing to the readers in our opinion.

58) ll 416 – "the accompanying material" instead of "associated supplementary material"

Reply: Corrected.

59) lls 426 – 428 This information does not belong in the conclusions.

Reply: The information was moved out of the conclusion.

60) ll 487 – "figure caption"

Reply: corrected.

61) Figs. B5, B11 – max. hits per layer must be 240

Reply: The max. hit per layer values of those diagrams are correct. It just so happens that some ceilometers (3 in Fig. B5, 1 in Fig. B11) acquired some data exactly at Delta t = 0 & 900 seconds. We use a selection criteria of <= 900 s for hits, hence the resulting max. hit per layer values of these figures.

```
**-----------------------------------------------------------**
**Reviewer 2**
**Primary comments**
```

This article presents a new algorithm: ampycloud, deployed by MeteoSwiss at different aerodromes within the country. As the authors state, given the opaqueness of ceilometer manufacturers' cloud layer height and amount algorithms, using these at a given aviation meteorological institute remains difficult. The article makes a cloud layer algorithm available, along with a suite of test cases for interested users to apply. The author present test cases and statistical studies in order to validate their algorithm. These, together with the algorithm deserve publication. However, changes in the manuscript must be made in order to improve its clarity. Specifically, more explanation is needed for certain parts. For example, the algorithm is tunable through many parameters, yet, the authors do not explain the reason for their choices of various constants. Also, their choice of integration time is not well argued and it is not immediately clear why vertical visibility is not reported with ampycloud. These comments are detailed in the following sections.

Reply: The article text has been adjusted throughout, to enhance clarity and avoid misunderstandings. The discussion of the different limitations of the code (e.g. in terms of VV handling) and the choices made by MeteoSwiss (e.g. 15min integration time) has also been improved.

**General comments**

1) Line 32: Please define auto metreport before using this name

Reply: Corrected.

2) Line 55–56: It also prevents combining different types of ceilometers. Combining different types of ceilometers might be done for e.g. financial reasons.

Reply: Agreed – this point was added to the text.

3) Introduction or section2: Some background reasoning or framing on why ampycloud is designed the way it is–> It uses rather advanced statistical methods compared to for example the Larsson algorithm used by the KNMI (see e.g. Wauben et al. 2002). What are the advantages/disadvantages of ampycloud compared to e.g. the algorithm described by Wauben?

Reply: We cannot speak about specific advantages/disadvantages of ampycloud over other similar algorithms, on the basis that we have not conducted any dedicated comparison (see our reply to point 27 for more details on why that is). We note however that ampycloud was not created because other algorithms are not good enough. Rather, it is the lack of detailed information about these algorithms that motivated MeteoSwiss to assemble its own, also to ensure its maintainability and extendability. The point is now mentioned in Sec. 2.1. We also note that we have been unable to locate the Larsson and Esbjörn (1995) reference quoted by Wauben (2002) as the source for the KNMI algorithm, and this even by contacting the Swedish Meteorological and Hydrological Institute (SHMI).

Regarding the complexity of ampycloud over other algorithms, we refer to our reply to point 2 of reviewer 1 (in short, other existing algorithms may not all be as simple as they appear to be on the surface).

4) line 94: I find (pitfall 1) difficult to understand

Reply: The word "thick" has been replaced with "span a broad range of heights" for clarity. See also our reply to point 32 below.

5) Line 103: "Obscuration of the sky by fog or snow must be handled by a separate algorithm". Please clarify as to why this is the case.

Reply: This sentence has been reworded to make it clear this is not a necessity per se, but simply how this is being done at MeteoSwiss. Our vertical visibility algorithm also combines horizontal visibiltiy and present weather measurements, which go beyond the scope of the ampycloud package.

6) Line 111: "relies on ceilometer hits". Perhaps a brief description of what a ceilometer hit is, together with a few words on how a ceilometer calculates cloud base heights would help the reader.

Reply: The term "ceilometer hit" has been replaced by "cloud and VV hits" throughout the article. The following text was added at the start of Sec. 2.3:

"The World Meteorological Organization (WMO) defines a cloud base as "a zone in which the obscuration corresponding to a change from clear air or haze to water droplets or ice crystals causes significant changes in the profiles of the backscatter and extinction coefficients'' (WMO, 2021). Ceilometers deployed at aerodromes worldwide are designed to detect such changes in backscatter coefficient profiles (ICAO, 2011), that are typically reported in the form of cloud base and/or vertical visibility (VV) hits (see e.g. Vaisala Oyj 2015, Campbell Scientific 2021, OTTH ydroMet 2022), but with significant differences between ceilometer types/models (Martucci et al., 2010; Görsdorf et al., 2016, 2018)""

7) Line 120: what is meant by relative time (It is described later, but perhaps better to describe it here)?

Reply: The sentence has been reworded as follows:

"the observation time (in seconds, either as absolute Unix time or relative to a specific event, e.g. the most recent ceilometer measurement)"

8) Line 123–125: "One should note that for the moment, ampycloud cannot use full back–scatter profiles to derive cloud base hits independently from the ceilometers' proprietary software." Since deriving your 'hits' from backscatter profiles would just mean that you're clustering different cloud base hits, does this affect the quality of the algorithm?

Reply: No, it does not. This point was clarified in the text with the following sentence:

"Implementing such a capability would help enhance the robustness and traceability of the hit derivation, but it would not fundamentally change the behavior of the \textsf{ampycloud} algorithm in itself."

9) Line 130: definition of AUTO METREPORT should be done earlier e.g. in the introduction

Reply: Agreed and done.

10) Line 132: Discussion Figure 2 should be in the main text.

Reply: Agreed and done.

11) Line 134–135: "The specification of hcid for every hit is necessary to compute a correct estimation of the sky covering fraction under the assumption that cloud layers have a unique base per time step per ceilometer line–of–sight.". Could you explain this further? Why is the hcid necessary for the correct estimation of the sky covering fraction given that all hits are treated equally, pooled together and sorted?

Reply: The following clarifying sentence has been added to the text: "Without a value of $h_{\rm cid}$ to differentiate sightlines, two simultaneous measurements from distinct ceilometers would not be distinguishable, and possibly contradictory."

12) Line 149: a delta t of 15 and 6 minutes seems arbitrarily chosen: Could you provide more details as to why 15 and 6 minutes were chosen?

Reply: The use of these time intervals stems from historical reasons. These intervals have been used for years at the Swiss airports, and our stakeholders are used to/comfortable with the resulting responsiveness of the system. We stress however that it is entirely up to the user to decide what pooling interval they wish to use – ampycloud will simply process the cloud hits that it is being fed. These points were clarified in the text.

13) Line 155: I was under the impression that compliant METAR cloud layer codes were derived sequentially, meaning that first S, then G then L were computed in order to finally give up to 3 layers. With the current wording, it seems that a METAR code is generated for the 'S step', the 'G step' and the 'L step'. Could you please clarify?

Reply: The reviewer is correct in that the three steps are required to derive the final answer. However, the outcome of each step also leads to a series of cloud "elements" (i.e. slices, groups, or layers). Each of these can be converted into a METAR code to help with the interpretation of what happens at each step, as illustrated in Fig. 1 (now Fig. 2). The sentence has been adjusted to avoid misunderstandings.

14) Line 156: Does that mean that a cloud layer is considered at the 4 hit? Does that mean that the cloud information other than NCD would be available to the AUTO METAR generation after 1 minute of detecting the beginning of a cloud layer (given a time resolution of 15 seconds for the ceilometer)? Could you say a few words on the operational suitability of this?

Reply: Yes, this is correct: 4 hits at the very end of the time interval would lead to the report of a cloud layer. In our experience, the measurements from ceilometers at LSGG and LSZH are sufficiently reliable that the detection of 4 hits does warrant the reporting of an associated cloud layer. It is however clear that a minimum of 4 hits may not necessarily be the optimal everywhere/with every ceilometer, e.g. if they measure at lower temporal resolution (e.g. 1 min). The following note was added at the start of Sec. 2.4:

"It must be stressed that the default values provided in Table 2 for the different parameters are not necessarily applicable universally. Depending on their needs and/or instrumental setups, some users may wish to consider adjusting some of them, for example Theta_0 and/or Theta_8."

15) Line 170–172: "If a user were to prefer the cloud base heights to be more representative of the most recent hits within a slice/group/layer, the B_t parameter can be set to lower values, e.g. B_t = 30%. Note that changing the value of _t does not have any effect on the cloud amounts." As far as I understand, this is not the same as doubly weighing the most recent cloud hits. When doubling the most recent cloud hits, the oldest cloud hits still have a single weight, and so are still considered in an algorithm (e.g. Larsson's algorithm). In ampycloud, setting B to 30% equates to only using 30% of the data within a slice. Is this correct? If not, could you please add a few words to clarify?

Reply: This is correct. We stress that beta_t only applies to the derivation of the base height value. As quoted by the reviewer, the value of beta_t does not have any impact on the sky coverage fractions (i.e. all the hits still contribute). While doubling the most recent cloud hits works quite well with histogram-based cloud algorithms, it cannot be transposed directly to the ampycloud clustering framework, where clusters are (mostly) defined based on distances between hits.

16) Line 164–165: "as commonly done by other cloud height algorithms" Are there any references to this?

Reply: In view of the difficulty to ascertain the exact behavior of other algorithms, this sentence has been removed from the text (without loss of meaning).

17) Line 209: why is alpha_s chosen as 0.2?

Reply: The value of 1/alpha_s indicates how many slices at most can be identified by ampycloud. With 1/alpha_s=5, we allow ampycloud some reasonable flexibility to identify numerous cloud layers in complex situations, without risking a significant over-slicing in "simple" cases. The text has been clarified accordingly.

18) Line 220: perhaps refer again to the over-slicing as the top layer of figure 1 to add clarity

Reply: Agreed and done.

19) Line 229: Please give a reason for choosing epsilon as 10%

Reply: This comment led us to realize that Eqs. (2) and (3) were wrong, and they have now been corrected. Essentially, we rely on the min and max hit heights to identify overlaps, with epsilon a small padding set to avoid edge cases (where the min and max heights have the same numerical values), and account for natural cloud base height fluctuations. This information has been added to the text.

20) Line 245: how is fb chosen?

Reply: The values |h–L| in Eq. 4 correspond to half(-ish) the group thickness, such that these need to be multiplied by two to obtain a suitable rescaling-height (i.e. to have the rescaled groups span a 0–1 range). Essentially, fb is not a parameter, but a part of our fluffiness definition. For clarity, we have therefore replaced fb by 2 in Eq. (4), and no longer mention this as a parameter of ampycloud. The Python package has been updated accordingly.

21) Line: 250 how is Lfrac chosen?

Reply: Lfrac determines the fraction of closest cloud hits used to derive the LOWESS fit at any location of a slice. The LOWESS fit will therefore be fully sensitive to height fluctuations affecting more than Lfrac*100% of the cloud hit sample. The smaller below Lfrac*100% the duration of a specific (coherent) height variation, the smaller its influence on the resulting LOWESS fit. We adopt Lfrac=0.35, which corresponds to a "sensitivity timescale" of ~5 min for uniform cloud hit distributions over an interval of 15 min. The article text has been clarified accordingly.

22) Line 330: Could you please give more information on how a human observer collects/validates data for his METAR: they derive a cloud layer height and amount based on the clouds in an area above and in the vicinity of the aerodrome. But do they also use raw ceilometer cloud hits? Do they use the results of another algorithm? Are they trained to more or less estimate cloud heights without ceilometer information?

Reply: Observers mostly estimate the cloud amount without using ceilometer data. Howerver, they are encouraged to use ceilometer measurements and also algorithm output for the estimation of cloud height, although the actual usage depends on the weather situation and preference of a given observer. The raw ceilometer measurements are plotted on a realtime display. This point has been added in Sec 3.4.

23) Line 332: how were the cases selected? Did you only select cases where human observers reported something operationally significant? Or did you select cases where either a human or the algorithm detected something important? Or did you select cases where both the algorithm and the human observers agreed that something of operational significance was occurred? Or was there another criteria used? Please specify as this could have consequences on the number of misses and false alarms. Also, can you say if using a larger sample (for example by using the whole 5 years) would affect the statistics?

Reply: The cases are selected when either the human observer or the algorithm (or both) reported something important, precisely to ensure that the sample covers misses and false alarms in a balanced way. This has been clarified in the text. Regarding the question of a large sample, we refer to our reply to point 6–7) in the other review.

24) Line 336: What is meant with operationally significant cloud events: are these ceilings? Low clouds? Low ceilings? Situations where at least 2 layers are present with at least a SCT in the second layer? And since these represent such a small sample compared to the whole 5 year dataset, does that mean that ampycloud is designed to be most accurate for operationally significant events at the expense of other events? Please clarify this.

Reply: "Operationally significant" refers to the presence of a ceiling, and the text has been clarified accordingly. This sample of cases was used for validation purposes, and not for fine-tuning the algorithm (the development/tuning was entirely done using the specific cases presented in the Appendix). From that perspective, the size of the sample does not affect the accuracy of the algorithm – it simply allows to evalute it.

25) Line 338–344: what is meant by a density of a cloud? Is it the number of oktas?

Reply: Yes. The term "density" has been replaced throughout the article.

26) Line 346: lines 333–336 seem to suggest that the 2128 cases are the high impact cases. In line 346 a further selection is made? Could you please clarify this?

Reply: No further selection is made, the matrix view has only been simplified. The confusing sentence has been replaced with: "The same information can be also be presented using operationally-relevant categories."

27) Line 367–369: "Understanding the exact behavioral differences between ampycloud and other algorithms with similar purpose would require a dedicated comparison of their respective performances against a reference slices, which is outside the scope of this article." Please provide more detail as to why such an analysis is not done. It is true that algorithms from manufacturers are kept secret, but in the case of Vaisala it is possible (against payment) to have the outcome of their algorithm in terms of cloud layer heights and cloud amount. Could you explain why a comparison of this output against ampycloud is not done? Furthermore, what is meant with a reference data set in this case?

Reply: "reference dataset" has been replaced with "reference set of ceilometer data (either real, or simulated)".

As far as we understand, the Vaisala software mentioned by the reviewer is not able to combine information from multiple instruments, i.e. it only works on a ceilometer-by-ceilometer basis. Furthermore, this software option has not been deployed at LSGG and LSZH, and is not available to us at this time.

The ampycloud accuracy was in fact compared to the cloud algorithm previously implemented within SMART. This comparison revealed that ampycloud does perform better than its predecessor (statistically speaking), with improved stability in case of complex situations. We however do not dicuss this comparison in the present article, as the original cloud algorithm implemented within SMART is neither publicly available nor publicly documented, such that this comparison would essentially be meaningless to the reader (even though it evidently was meaningful to MeteoSwiss and the Swiss Civil Aviation Authority). Only a comparison against a public cloud algorithm would be meaningful in the present setting. We absolutely agree that this is very much warranted, but remain of the opinion that the black-box/unpublished/opaque nature of the other algorithms place this perspective well outside the scope of this article.

28) Line 415–417: Should this be in the results section?

Reply: Agreed and done.

29) Line 419–429: In the method section?

Reply: Agreed, but moved to Sec. 4 (limitations).

30) Figure B10: low visibility procedures should be introduced

Reply: There is no need to discuss Low Visibility Procedures in this caption, and we have removed the sentence entirely (without loss of meaning).

31) Figure B12: information about VV being decided by a separate algorithm should be in the main text in the introduction or methodology section, together as to why meteoswiss/the authors decided this that way. Also, as VV is not within ampycloud, it should be emphasized that ampycloud is only partly compliant with ICAO's rules for cloud reporting.

Reply: The topic of VV reporting is discussed (now in an improved manner) in Sec. 2, where we have also added the explicit mention that "... when used on its own, \textsf{ampycloud} can comply with only a subset of the ICAO's rules for cloud reporting, ...". We also refer to point 2 of reviewer 1, where we explain that even when used together with a VV algorithm, we still would not have sufficient information to determine the codes NSC and CAVOK.

**Minor comments**

31) Please use 'inputed' instead of 'fed'

Reply: Corrected throughout.

32) Lines 94–95: starting the sentences with 'It' may make the sentences more readable

Reply: Agreed and done.

33) Lines 114–115: Better to write: "The longer the time interval or the larger the wind speed, the better the spatial representativity of the dataset, but the worse is the view of the current state of the sky"

Reply: Agreed and done.

34) Line 127: use 'apply' instead of 'run'

Reply: Agreed and implemented.

35) Line 353–355: "But it cannot guarantee that the "correct" cloud layers are being identified when they have the correct density." Perhaps better to say that detecting the right number of oktas for a cloud layer cannot guarantee that the cloud layer has the correct height?

Reply: Agreed and implemented.

36) Line 390: small amount of information implies that you are talking about a small number of cloud hits, which is not the case. In this sentence, please consider saying that it relies on cloud hits only.

Reply: Agreed and done.

37) Line 399: use the word "detect" instead of "see"

Reply: Agreed and done.

38) Line 489: Please define "scientific stability"

Reply: "assess the scientfic stability" has been replaced with "monitor the scientific behavior".

---

## Referee Report (RR1)

**Review of the manuscript 'ampycloud: an open-source algorithm to determine cloud base heights and sky coverage fractions from ceilometer data'**

Peter Kuma

Department of Meteorology (MISU), Stockholm University, Stockholm SE-106 91, Sweden

15 June 2024

Dear Editor and Authors,

Here I present only a high-level review of the manuscript as requested by the editor. First, I want to disclose that I am not an expert on aviation rules and regulations, and I can only comment on the atmospheric physics aspects of the manuscript. I find the proposed algorithm reasonable and the authors' presentation satisfactory, although some points are unnecessarily convoluted. The biggest shortcoming is the reliance on the instrument's determined cloud base and vertical visibility. The authors present the algorithm as a way of avoiding the black box algorithms of ceilometers, but their algorithm is still crucially dependent on the black box algorithm used in the Vaisala CL31. There is no guarantee that different current or future models will use the same underlying algorithm, and this can in turn make the authors' algorithm unreliable depending on which particular CL31 models are used (I recommend reading Kotthaus et al., 2016 on this matter). It would obviously be much better to base the proposed algorithm on the backscatter reported by the instrument. This would also allow for more complex processing because the backscatter provides information about cloud thickness and can measure multiple cloud layers in one profile, to the extent that the laser signal is not attenuated. That said, if the potential users of the algorithm are warned about this and perform testing on their set of ceilometers before operational use, it might not be a big issue. I cannot judge the compliance with ICAO rules, and Referee 1 might have valid concerns on this. I also agree with Referee 1 that the methods used are perhaps too complex for the task. At the same time, performance is not really an issue with this algorithm because it is sufficiently fast for operational use. Another smaller issue is that the algorithm description (slicing, grouping, and layering) is relatively complicated, and the manuscript would benefit from a preceding section describing all three steps and the motivation for them in a high-level overview, as well as a better description of the parameters. For example, by adding a one-sentence description of each in Table 2. Overall, I am inclined to recommend the publication of this manuscript.

Kind regards,

Peter Kuma

**References**

Kotthaus, S., O'Connor, E., Münkel, C., Charlton-Perez, C., Haeffelin, M., Gabey, A. M., and Grimmond, C. S. B.: Recommendations for processing atmospheric attenuated backscatter profiles from Vaisala CL31 ceilometers, Atmos. Meas. Tech., 9, 3769–3791, https://doi.org/10.5194/amt-9-3769-2016, 2016.

---

## Author Response (AR2)

We thank the anonymous reviewer #2 and Peter Kuma for their feedback on the updated article. As suggested, we have shortened the caption of Fig. 2 by moving it into a dedicated Section. The (current) dependence on the black-box algorithms from manufacturer ceilometers has also been made explicit in the introduction, conclusions and in Sec. 4. Our detailed replies to the different suggestions are provided below.

The authors
July 2024

**----------------------------------------------------------------**
**Editor**

1) The authors carefully replied to the comments of the two reviewers made in the discussion phase. The manuscript was adapted accordingly.

With respect to the general criticism of reviewer 1, the authors plausible outlined their approach and philosophy, explained the position of ampycloud within the full processing chain of SMART and addressed the limitations of ampycloud more clearly. Therefore, I see most of the criticism removed. To my understanding, ampycloud might not be the final and perfect solution for cloud detection with ceilometers, but also small contributions to the scientific discussion are helpful. Thus, I also honor the approach of the authors to make the algorithm publicly available. Their tool might help other researchers using ceilometer measurements and lead to further development of cloud detection algorithms, as already announced in their replies.

An additional reviewer commented on the revised version of the manuscript after the discussion phase and raised minor issues. The authors should consider these issues in another revision of the manuscript. Especially the crucial dependence of ampycloud on the black box algorithm used in the Vaisala CL31 should be discussed in more depths. I recommend to outline a strategy to adapt ampycloud for next generation ceilometer in future.

Reply: The fact that ampycloud relies on cloud base hits reported by ceilometers (via black-box algorithms) is now written explicitly in the abstract of the article and in the conclusion. The Kotthaus (2016) reference has been added to Sec. 4 (where the reliance on cloud base hits is being discussed) together with the following clarifying sentence:

"This (current) reliance of \textsf{ampycloud} on cloud base hits (as reported by ceilometers via black-box algorithms) should not be overlooked by users interested to combine datasets from multiple ceilometer brands/ models."

We have also updated the package documentation (i.e. the webpages hosted on Github describing how to use the ampycloud Python package) to stress the dependence of the code on the ceilometer algorithms.

2) I finally recommend to shorten the caption of Fig. 2. This hardly will work in the final journal layout. The discussion of figures should be made in the main text. For this specific figure you could either provide and describe the ampycloud output screen in figure section 2.4 or discuss the figure as one example in an extra section. The current version of the caption is also hard to understand and orientation is difficult, e.g. I'm lost for "The final selection of cloud layers is shown in the top right of the diagram. To the bottom left, the number of ceilometers contributing data to the diagram is indicated...". That's why it would be OK to just have a caption like "Example of ampycloud. Details are explained in Section...." and then carefully guide through the individual parts in the main text.

Reply:

The description of Fig.2 is now made inside the (new) Section 2.5, with the Figure caption simply referring to it for details.

**----------------------------------------------------------------**
**Referee 3**

1) Dear Editor and Authors,
Here I present only a high-level review of the manuscript as requested by the editor. First, I want to disclose that I am not an expert on aviation rules and regulations, and I can only comment on the atmospheric physics aspects of the manuscript. I find the proposed algorithm reasonable and the authors' presentation satisfactory, although some points are unnecessarily convoluted. The biggest shortcoming is the reliance on the instrument's determined cloud base and vertical visibility. The authors present the algorithm as a way of avoiding the black box algorithms of ceilometers, but their algorithm is still crucially dependent on the black box algorithm used in the Vaisala CL31. There is no guarantee that different current or future models will use the same underlying algorithm, and this can in turn make the authors' algorithm unreliable depending on which particular CL31 models are used (I recommend reading Kotthaus et al., 2016 on this matter). It would obviously be much better to base the proposed algorithm on the backscatter reported by the instrument. This would also allow for more complex processing because the backscatter provides information about cloud thickness and can measure multiple cloud layers in one profile, to the extent that the laser signal is not attenuated. That said, if the potential users of the algorithm are warned about this and perform testing on their set of ceilometers before operational use, it might not be a big issue.

Reply: See our reply to the Editor comment #1.

2) I cannot judge the compliance with ICAO rules, and Referee 1 might have valid concerns on this. I also agree with Referee 1 that the methods used are perhaps too complex for the task. At the same time, performance is not really an issue with this algorithm because it is sufficiently fast for operational use.
Another smaller issue is that the algorithm description (slicing, grouping, and layering) is relatively complicated,
and the manuscript would benefit from a preceding section describing all three steps and the motivation for them in a high-level overview, as well as a better description of the parameters. For example, by adding a one-sentence description of each in Table 2. Overall, I am inclined to recommend the publication of this manuscript.

Reply:

A high-level description of the 3 algorithm steps, and their inter-dependence, is provided at the start of Sec. 2.2. Following the referee's comment, we considered moving it at the start of Sec. 2.6, but eventually opted to

keep it in its current position (which we think offers a better flow to the reader).

Regarding the suggestion of adding a brief sentence for each parameter in Table 2, we remain of the opinion that doing so would complicate the Table while not being necessarily easier to understand. In particular, not every parameter can be easily understood with a single, short sentence in isolation. Instead, having the link to the specific Section where each parameter is defined should allow readers to easily locate the description (in context) of each item in Table 2.